# Age influences on the molecular presentation of tumours

Constance H. Li[1,2,3,4,5,6], Syed Haider [7] & Paul C. Boutros [2,3,4,5,6,8,9 ✉]

Cancer is often called a disease of aging. There are numerous ways in which cancer epidemiology and behaviour change with the age of the patient. The molecular bases for these relationships remain largely underexplored. To characterise them, we analyse age-associations in the nuclear and mitochondrial somatic mutational landscape of 20,033 tumours across 35 tumour-types. Age influences both the number of mutations in a tumour (0.077 mutations per megabase per year) and their evolutionary timing. Specific mutational signatures are associated with age, reflecting differences in exogenous and endogenous oncogenic processes such as a greater influence of tobacco use in the tumours of younger patients, but higher activity of DNA damage repair signatures in those of older patients. We find that known cancer driver genes such as *CDKN2A* and *CREBBP* are mutated in age-associated frequencies, and these alter the transcriptome and predict for clinical outcomes. These effects are most striking in brain cancers where alterations like *SUFU* loss and *ATRX* mutation are age-dependent prognostic biomarkers. Using three cancer datasets, we show that age shapes the somatic mutational landscape of cancer, with clinical implications.

[1] Ontario Institute for Cancer Research, Toronto, ON, Canada. [2] Department of Medical Biophysics, University of Toronto, Toronto, ON, Canada. [3] Department of Human Genetics, University of California, Los Angeles, CA, USA. [4] Department of Urology, University of California, Los Angeles, CA, USA. [5] Jonsson Comprehensive Cancer Center, University of California, Los Angeles, CA, USA. [6] Institute for Precision Health, University of California, Los Angeles, CA, USA. [7] The Breast Cancer Now Toby Robins Research Centre, The Institute of Cancer Research, London, UK. [8] Department of Pharmacology & Toxicology, University of Toronto, Toronto, ON, Canada. [9] Vector Institute for Artificial Intelligence, Toronto, ON, Canada. ✉email: PBoutros@mednet.ucla.edu

Cancer health disparities across different population stratifiers are common through a wide range of measures. These include differences in incidence rates, mortality rates, response to treatment, and survival between individuals of different sexes[1–6], races or ancestries[7–11] and ages[12–14], and these differences have been described across a range of tumour-types. Cancer disparities involving age are particularly well-known. Aging is a leading risk factor for cancer, as it is associated with increased incidence of most tumour-types[9,15]. Older age is also typically associated with higher mortality and lower survival[16,17]. The links between older age and increased cancer burden are such that cancer is often described as a disease of aging[18,19].

However, there are many nuances in the relationship between aging and cancer. Paediatric cancers are an obvious exception, as cancers arising in children have different molecular and clinical characteristics[9,20–22]. Tumours arising in young adults (<50 years of age) are often more aggressive: early onset tumours of the prostate[23], breast[24], pancreatic[25,26], colorectal[27] and soft tissue sarcomas[28] are diagnosed at higher stages and associated with lower survival. Molecular studies have described some striking differences in the mutational landscapes of early onset vs. later onset disease[29–31], suggesting differences in the underlying oncogenic processes driving cancer at different ages.

The mechanisms of how age shapes the clinical behaviour of cancers have been subject to intense study. Many factors and behaviours closely tied to aging have been implicated in observed epidemiological and clinical cancer health disparities. For example, higher age is associated with a greater burden of comorbidities such as diabetes and cardiovascular disease[32,33]. Higher prevalence of chronic disease, frailty and increased likelihood of adverse drug reactions also influence the choices of clinical interventions given to older cancer patients[34–36]. Nevertheless differences remain even after accounting for these factors[37]. Previous work associating somatic molecular changes with age suggest differences in overall tumour mutation burden[38], transcriptional profiles[39] and some mutational differences[29–31]. These studies have focused on single tumour-types, relatively small cohorts, or have only evaluated fractions of the whole-genome, leaving the landscape of age-associated cancer mutations largely unknown.

In this work, we perform a pan-cancer, genome-wide study of age-associated molecular differences in 10,218 tumours of 23 tumour-types from The Cancer Genome Atlas (TCGA), 2562 tumours of 30 tumour-types from the International Cancer Genome Consortium/TCGA Pan-cancer Analysis of Whole Genomes (PCAWG) and 7259 tumours across 35 tumour-types from the AACR Genomics Evidence Neoplasia Information Exchange (GENIE) projects. We quantify age-associations in measures of mutation density, subclonal architecture, mutation timing, mutational signatures and driver mutations in almost all tumour-types. These associations remain even after adjusting for potential confounding factors such as sex and ancestry. Many of these genomic age-associations are linked to clinical phenotypes: in particular, we identify genomic alterations that are prognostic in specific age contexts, suggesting the clinical utility of age-informed biomarkers.

## Results

**Age associations in mutation density and timing.** We investigated TCGA, PCAWG and AACR GENIE datasets independently and performed pan-cancer analyses spanning all TCGA, all PCAWG, and all AACR GENIE tumours in separate analyses; these were supplemented with tumour-type-specific analyses. We used the recorded age at diagnosis (Table 1) and implemented a two-stage statistical approach: we first used univariate methods to identify putative age-associations, then further modeled these putative hits with multivariate regression to evaluate age effects after adjusting for confounding factors. Our multivariate modeling accounted for a range of confounding variables for each tumour-type including sex and genetic ancestry. We modeled each genomic feature and tumour subtype based on available clinical data, a priori knowledge, variable collinearity and model convergence. Model and variable specifications, and results of association tests between model variables and age are presented in Supplementary Data 1. We performed two rounds of multiple testing adjustment: once at the first univariate stage, and again at the second multivariate stage, both using the Benjamini–Hochberg false discovery rate (FDR) procedure. Our findings must pass stacked FDR thresholds of 10% on top of 10% after both stages of analysis, representing a stringent combined threshold of 1%. Bonferroni-adjusted $p$ values provided similar support for our findings. FDR-adjusted $p$ values were reported unless otherwise noted. Both Benjamini–Hochberg and Bonferroni, as well as unadjusted $p$ values are provided in supplementary materials. We present the subset of statistically significant results in Supplementary Data 2, and full results in Supplementary Data 3–7.

The accumulation of mutations with age is a well-known phenomenon in both cancer and non-cancer cells[40–47]. To test the robustness of our statistical framework in detecting age-associated genomic events, we investigated age associations in two measures of mutation accumulation: single nucleotide variant (SNV) density and genome instability. Both SNV density and genome instability have clinical relevance as they are associated with poor outcome in some tumour-types[48–50] and response to immunotherapy in others[51,52]. We first identified univariate age-associations in SNV density using Spearman correlation. Putative age-associations identified at an FDR threshold of 10% were further analysed by multivariate linear regression (LNR) models to adjust for tumour-type-specific confounding effects (Supplementary Data 1) and a second FDR threshold of 10% was used to identify statistically significant age-associated events. We have previously applied this statistical strategy successfully to identify sex-associated somatic mutational features[53,54].

As expected[40,41], we identified significant associations between age and the accumulation of SNVs across a range of tumour contexts (Fig. 1A). There were pan-cancer positive correlations between age and SNV density in TCGA (pan-TCGA: $\rho = 0.31$, FDR-adjusted LNR $p = 4.1 \times 10^{-57}$, Bonferroni-adjusted LNR $p = 4.1 \times 10^{-57}$) and PCAWG ($\rho = 0.43$, FDR-adjusted LNR $p = 1.6 \times 10^{-26}$, Bonferroni-adjusted LNR $p = 4.1 \times 10^{-57}$) data. Using TCGA and PCAWG data, we estimated that SNV density increases at a rate of 0.077 mutations per megabase pair per year (Table 2, *Methods*). We also identified positive associations in 11 TCGA, 14 PCAWG and six AACR GENIE tumour-types (Fig. 1A). Of these, nine tumour-types showed consistent results in two of three datasets (Supplementary Fig. 1, Supplementary Data 2, Supplementary Data 3) including prostate cancer (TCGA: $\rho = 0.25$, FDR-adjusted LNR $p = 0.015$, Bonferroni-adjusted LNR $p = 0.13$; PCAWG: $\rho = 0.48$, FDR-adjusted LNR $p = 1.2 \times 10^{-4}$, Bonferroni-adjusted LNR $p = 8.7 \times 10^{-4}$, estimated 0.12 mut/Mbp/year; Fig. 1B). Estimates for per year increase in mutation density are given in Table 2 for the nine tumour-types with consistent evidence in at least two datasets.

We next asked whether there were differences in the timing of when these SNVs occurred during tumour evolution and leveraged data describing the evolutionary history of PCAWG tumours[55]. We first investigated polyclonality, or the number of cancer cell populations detected in each tumour. Monoclonal tumours, or those where all tumour cells are derived from one ancestral cell, are associated with better survival in several

**Table 1 Summary of age data per tumour-type.**

| Tumour type | TCGA age | PCAWG age | GENIE-MSK age |
| --- | --- | --- | --- |
| Bladder Cancer | 34–90 (69) | 34–84 (65) | 26–90.1 (67) |
| Breast Carcinoma | 26–90 (59) | 30–89 (56) | 19–90.1 (55) |
| Biliary Cancer | – | 37–84 (64) | 24–85.1 (64) |
| Cervical Cancer | 20–88 (46) | 21–58 (39) | 26–72 (48) |
| Colorectal Cancer | 31–90 (68) | 31–89 (67.5) | 18–90.1 (56) |
| Glioblastoma | 21–89 (62) | 21–76 (59) | 18–90.1 (60) |
| Medulloblastoma | – | 18–49 (26) | 28–72 (46) |
| Pilocytic Astrocytoma | – | 20–50 (24) | 29–44 (36.5) |
| Head and Neck Carcinoma | 19–90 (61) | 19–76 (53) | 28–90.1 (59) |
| Clear Cell Renal Cell Carcinoma | 26–90 (61) | 38–84 (60) | 31–85.1 (60) |
| Chromophobe Renal cell Carcinoma | 26–86 (50) | 28–86 (47.5) | 30–73 (56) |
| Papillary Cell Renal Cell Carcinoma | 28–88 (61) | – | 36–77.1 (60) |
| Non-Hodgkin Lymphoma | – | 18–85 (62) | 29–84.1 (60) |
| Chronic Lymphocytic Leukaemia | – | 40–86 (61) | 44–72 (67) |
| Acute Myeloid Leukaemia | 18–88 (58) | 35–75 (50) | – |
| Myeloid-MPN | – | 27–85 (54) | – |
| Lower Grade Glioma | 18–87 (41) | 21–62 (42) | 21–81.1 (48) |
| Hepatocellular Carcinoma | 18–90 (61) | 23–89 (67) | 18–90.1 (63) |
| Lung Adenocarcinoma | 38–88 (67) | 41–81 (65.5) | 22–90.1 (68) |
| Lung Squamous Cell Carcinoma | 40–85 (68) | 47–83 (68) | 30–90.1 (68) |
| Ovarian Cancer | 26–89 (59) | 39–81 (60) | 26–84.1 (59) |
| Pancreatic Cancer | 35–85 (66) | 34–90 (67) | 32–90.1 (67) |
| Pancreatic Endocrine Cancer | – | 20–81 (59) | 36–85.1 (58) |
| Pheochromocytoma | 19–83 (46) | – | 26–59 (41.5) |
| Prostate Cancer | 41–78 (61) | 38–80 (59) | 43–90.1 (65) |
| Sarcoma | 20–90 (60) | – | 18–84.1 (52) |
| Melanoma | 18–87 (55.5) | 19–87 (57.5) | 24–90.1 (64) |
| Stomach Adenocarcinoma | 34–90 (67) | 36–90 (65) | 18–85.1 (59.5) |
| Esogapheal Carcinoma | 27–90 (61) | 44–87 (70) | 18–88.1 (59) |
| Thyroid Cancer | 18–89 (46) | 18–85 (51) | 22–88.1 (63) |
| Thymic Tumour | 31–84 (61) | – | 20–83.1 (59.5) |
| Endometrial Carcinoma | 33–90 (63) | 35–90 (69) | 42–90.1 (60) |
| *Pan-cancer* | 18–90 (60) | 18–90 (62) | 18–90.1 (60) |

tumours types[56–58]. While there were intriguing univariate associations between age and polyclonality in non-Hodgkin lymphoma and prostate cancer, these were not significant in multivariable modeling (Supplementary Fig. 1, Supplementary Data 3). We then focused on polyclonal tumours and asked whether there were associations in mutation timing: we investigated whether SNVs, indels or structural variants (SVs) occurred more frequently as clonal mutations in the trunk or as subclonal ones in branches.

We identified several significant associations between age and mutation timing. In pan-PCAWG analysis, we found positive associations between age and proportion of clonal SNVs ($\rho = 0.20$, FDR-adjusted LNR $p = 1.4 \times 10^{-3}$, Fig. 1C) and proportion of clonal indels ($\rho = 0.14$, LNR $p = 0.013$, Supplementary Data 3). Age was also associated with increasing clonal SNV proportion in two tumour-types: stomach cancer (Stomach-AdenoCA: $\rho = 0.44$, FDR-adjusted LNR $p = 0.028$, Bonferroni-adjusted LNR $p = 0.11$), and medulloblastoma (CNS-Medullo: $\rho = 0.34$, FDR-adjusted LNR $p = 2.5 \times 10^{-3}$, Bonferroni-adjusted LNR $p = 5.1 \times 10^{-3}$, Fig. 1C). Positive correlations in these tumour-types indicate tumours arising in older individuals accumulated a greater fraction of SNVs earlier in tumour evolution. In contrast, we identified the inverse trend in melanoma, where tumours of younger patients accumulated more subclonal than clonal SNVs ($\rho = -0.47$, FDR-adjusted LNR $p = 7.8 \times 10^{-3}$, Bonferroni-adjusted LNR $p = 0.023$). Differences in the proportion of clonal mutations suggest differential mutation timing over the tumour evolution and could be caused by mechanisms such as differences in mutational processes or driver mutation frequency.

We next focused on genome instability, a measure of copy number aberration (CNA) burden and approximated by the percent of the genome altered by CNAs (PGA). Analogous to SNV density measuring the burden of point mutations, PGA measures the density of copy number alterations. We found that in pan-cancer analysis, PGA increased with age in PCAWG ($\rho = 0.19$, FDR-adjusted LNR $p = 0.022$, Bonferroni-adjusted LNR $p = 0.068$) and AACR GENIE ($\rho = 0.041$, FDR-adjusted LNR $p = 0.050$, Bonferroni-adjusted LNR $p = 0.16$) (Fig. 1D) and estimate that PGA increased at 0.010% per year (Table 2). We also identified positive correlations in six TCGA, three PCAWG, and three AACR GENIE tumour-types. Again, prostate cancer showed consistent age-PGA associations, this time in all three datasets (TCGA: $\rho = 0.17$, FDR-adjusted LNR $p = 6.7 \times 10^{-5}$, Bonferroni-adjusted LNR $p = 1.8 \times 10^{-4}$; PCAWG: $\rho = 0.27$, FDR-adjusted LNR $p = 3.0 \times 10^{-3}$, Bonferroni-adjusted LNR $p = 4.4 \times 10^{-3}$; AACR GENIE: $\rho = 0.11$, FDR-adjusted LNR $p = 0.050$, Bonferroni-adjusted LNR $p = 0.20$, increase of 0.2%/year; Fig. 1E). Age was associated with PGA in stomach cancer data in TCGA with an estimated increase of 0.19% per year ($\rho = 0.11$, FDR-adjusted LNR $p = 0.011$, Bonferroni-adjusted LNR $p = 0.011$) and AACR GENIE ($\rho = 0.38$, FDR-adjusted LNR $p = 0.041$, Bonferroni-adjusted LNR $p = 0.083$), and while other age-PGA correlations were not statistically significant across multiple datasets, they showed similar effect sizes (Supplementary Fig. 1). Intriguingly, we detected negative age-PGA associations in TCGA lung adenocarcinomas (Fig. 1D), and correspondingly negative associations in PCAWG ($\rho = -0.13$) and AACR GENIE lung tumours ($\rho = -0.099$) (Supplementary Fig. 1). Estimates for per year increase in PGA are given in

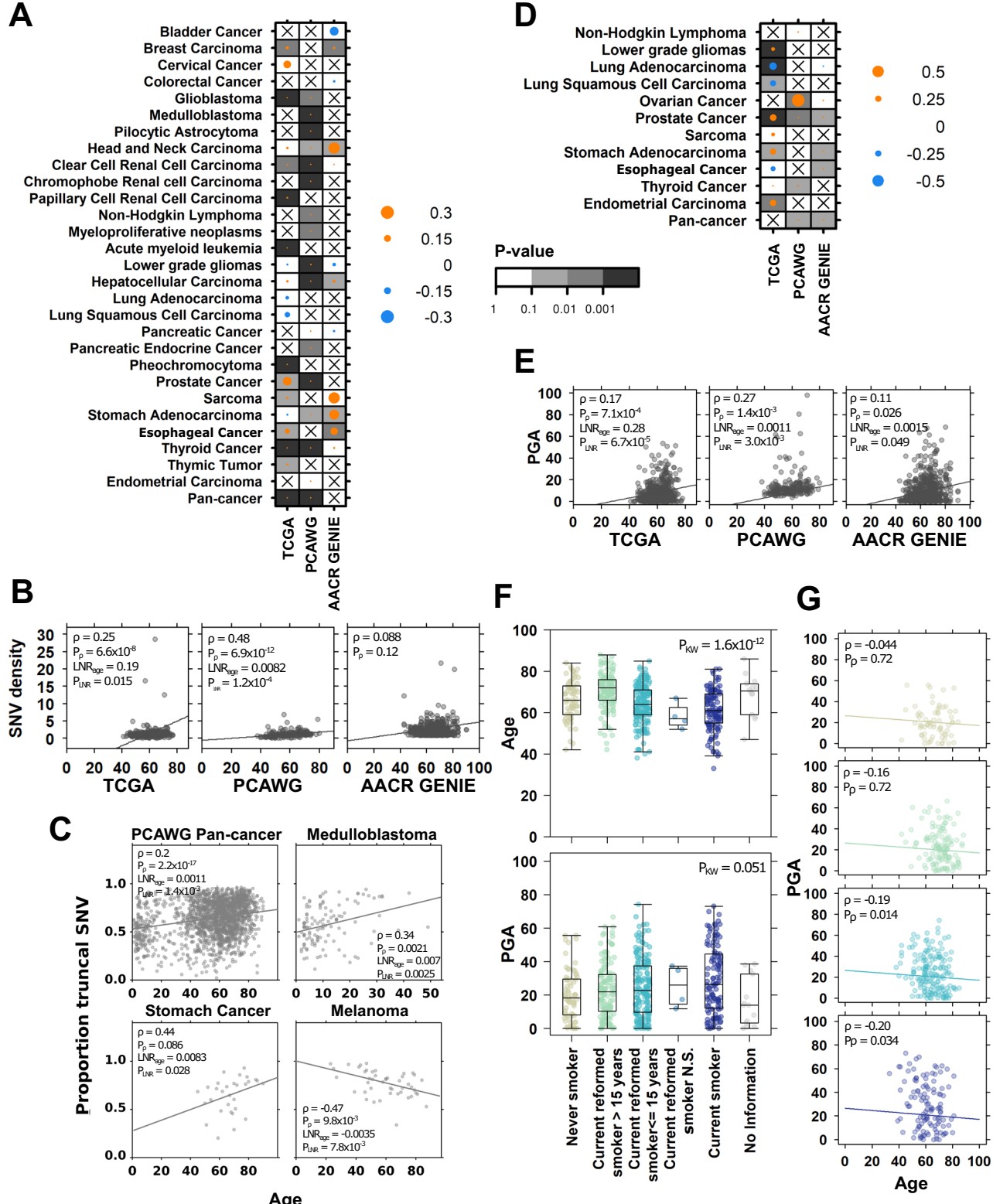

Table 2 for the five tumour-types with consistent evidence in at least two datasets.

To better understand why lung cancers of younger patients might have higher genomic instability, we investigated tobacco use. Tobacco exposure is a well-known risk factor in lung cancers. Our multivariable lung cancer regression models do account for tobacco use, but we supplemented our analysis with a more focused study of the relationship between PGA, age and tobacco

history. Age was associated with tobacco history (Kruskal–Wallis $p = 1.6 \times 10^{-12}$, Fig. 1F, *top*), where current smokers were diagnosed with lung cancer at younger ages than never and reformed smokers (difference in location $= -6.0$, 95% CI $= -8.0--4.0$, Wilcoxon rank sum test $p = 1.2 \times 10^{-7}$). PGA was also associated with smoking history (Kruskal–Wallis $p = 1.6 \times 10^{-12}$, Fig. 1F, *bottom*) where tumours arising in current smokers had higher PGA (difference in location $= 5.0$,

**Fig. 1 Mutation density and timing are associated with age at diagnosis.** Summary of associations between age and (**A**) SNV density and (**D**) percent genome altered (PGA) in TCGA, PCAWG and AACR GENIE tumours. The dot size and colour show the Spearman correlation, and background shading indicate adjusted multivariate $p$ value. Only tumour-types with at least univariately significant associations are shown. Associations between (**B**) SNV density and (**E**) PGA with age in prostate cancer across the three datasets ($n_{TCGA} = 492$, $n_{PCAWG} = 199$, $n_{AACR-GENE} = 582$ biologically independent samples). Univariate Spearman correlation, FDR-adjusted correlation $p$ value and FDR-adjusted multivariable linear regression $p$ values shown. **C** Correlations between age and proportion of SNVs occurring in the truncal clone in four PCAWG tumour contexts (Spearman correlation and linear regression FDR-adjusted $p$ values). **F** In TCGA lung adenocarcinoma ($n = 497$ biologically independent samples), age and PGA are associated with smoking history (Kruskal–Wallis test), and (**G**) the negative association between PGA and age remains significant in current smokers and recently reformed smokers (≤15 years; Spearman correlation $p$ values). From top to bottom: never smokers (yellow), current reformed smokers >15 years (green), current reformed smokers ≤15 years (light blue), current smokers (dark blue). Tukey boxplots are shown with the box indicating quartiles and the whiskers drawn at the lowest and highest points within 1.5 interquartile range of the lower and upper quartiles, respectively. Source data are provided as a Source Data file.

**Table 2 Estimates of per year increase in mutation density.**

| Tumour-type | ΔMut/Mbp per year (95% CI) | ΔPGA per year (95% CI) |
|---|---|---|
| Breast Carcinoma | 0.064 (0.029–0.95) | – |
| Glioblastoma | 0.018 (−0.020–0.056) | – |
| Head and Neck Carcinoma | 0.14 (0.071–0.21) | – |
| Clear Cell Renal Cell Carcinoma | 0.018 (0.0084–0.027) | – |
| Lung Adenocarcinoma | – | −0.17 (−0.24–0.10) |
| Hepatocellular Carcinoma | 0.069 (0.032–0.11) | – |
| Ovarian Cancer | – | 0.61 (0.41–0.81) |
| Prostate Cancer | 0.12 (0.034–0.20) | 0.2 (0.13–0.27) |
| Sarcoma | 0.044 (−0.0091–0.098) | – |
| Stomach Adenocarcinoma | 0.31 (0.14–0.49) | 0.19 (0.064–0.31) |
| Thyroid Cancer | 0.0082 (0.0066–0.0097) | 0.067 (0.036–0.098) |
| Pan-cancer | 0.077 (0.049–0.10) | 0.010 (−0.01–0.030) |

95% CI = 1.2–9.2, Wilcoxon rank sum test $p = 1.0 \times 10^{-3}$). We then examined the associations between age and PGA within each tobacco use category and found no association in never and reformed smokers ≥15 years (Spearman correlation $p \geq 0.1$, Fig. 1G). In contrast, tumours of current and reformed smokers <15 showed statistically significant negative correlations, indicating the negative association between age and PGA are dependent on current or recent tobacco use. These mutation density results demonstrate nuances in the relationship between aging and the accumulation of mutations: there are differences by when mutations occur throughout tumour evolution, and by how they occur through different mutational processes.

**Age-associated mutational signatures.** We continued exploring mutational processes through mutational signatures data. Clinical data on exposures such as tobacco use history and frequency are self-reported by the patient and may be inaccurate, misrecorded or not described. Moreover, clinical data often do not include information on exposures such as second-hand smoke, which is also a known cancer risk factor[59]. We leveraged the COSMIC mutational signatures, which can be applied to deconvolve distinctive mutational patterns from genomic sequencing[60]. Each mutational signature is thought to represent a specific oncogenic process—for instance, single base signature (SBS) 4 represents tobacco exposure. Using mutational signatures, we can not only detect the influence of oncogenic processes within a tumour, but we can also quantify the number of mutations attributed to that process and assess its relative activity compared with the activity of other mutational processes active within the tumour.

Focusing on the tobacco smoking signature SBS4, we stratified TCGA lung adenocarcinoma tumours into SBS4-postive (SBS4+) and -negative (SBS4−) groups. SBS4+ tumours were detected more frequently in younger patients (Wilcox $p = 1.1 \times 10^{-5}$) and had higher PGA (Wilcox $p = 1.0 \times 10^{-3}$, Fig. 2A left). We also

found that the number of mutations attributed to SBS4 was negatively correlated with age (Spearman's $\rho = -0.12$, $p$ value = 0.086) and positively correlated with PGA (Spearman's $\rho = 0.31$, $p$ value = $2.9 \times 10^{-6}$, Fig. 2A centre). These findings show that a greater burden of tobacco-associated mutations occurred in lung tumours of younger patients and was associated with increased genomic instability. When considering SBS4-attributed SNVs as a fraction of total detected SNVs, we found that while age remained negatively correlated with SBS4 relative activity (Spearman's $\rho = -0.16$, $p$ value = 0.017), there was no association with PGA (Spearman's $\rho = 0.082$, $p$ value = 0.23, Fig. 2A right): i.e. while smoking-attributed mutations accounted for a greater percentage of mutations in younger patients, this relative activity was not associated with a change genomic instability. Also, when examining the relationships between age and PGA separately in SBS4+ and SBS4− tumours, we found significant negative correlations in both groups (Fig. 2B), suggesting additional factors beyond SBS4-attributed mutations may contribute to increased PGA in lung tumours diagnosed in younger patients.

We repeated this analysis in TCGA lung squamous cell cancers and detected a similar negative correlation between age and PGA in SBS4- tumours (Spearman's $\rho = -0.17$, $p$ value = $5.8 \times 10^{-3}$, but no association in SBS+ tumours (Spearman's $\rho = -0.0069$, $p$ value = 0.94, Supplementary Fig. 2). SBS4 activity was also negatively associated with age in PCAWG lung adenocarcinomas (Lung-AdenoCA: $\rho = -0.50$, adjusted LNR $p = 0.025$, Supplementary Fig. 2). Indeed, SBS4 and age were consistently negatively associated across both subtypes of lung cancer and both datasets, though not all associations were statistically significant after multiple testing adjustment. This supports previous findings that tobacco has a larger tumorigenesis role in younger patients, with tobacco-associated mutations contributing to a greater portion of the mutational landscape of tumours derived from younger individuals[61].

The PCAWG project updated the COSMIC signatures v3 to a set of 49 single base substitution, 11 doublet base substitution

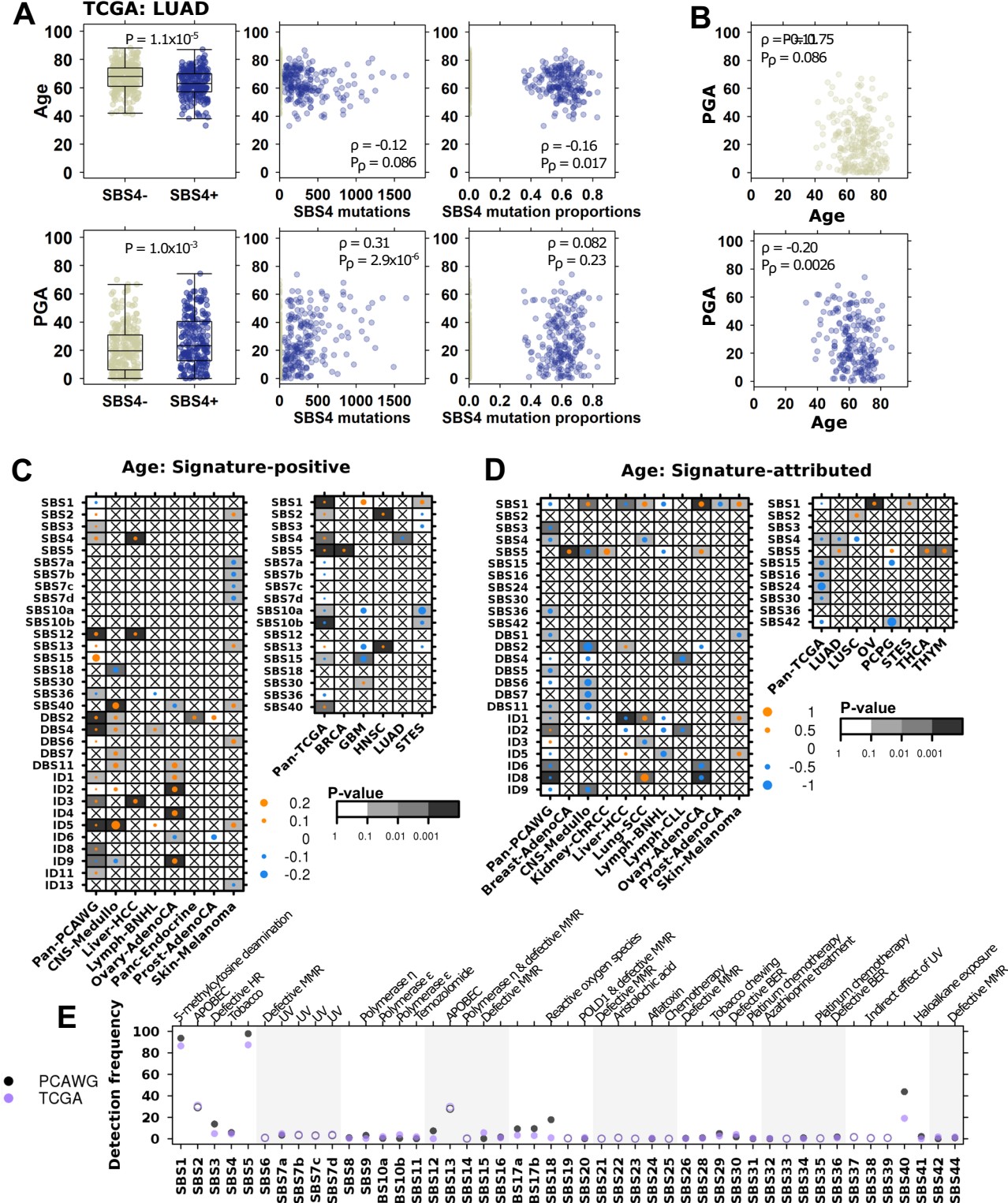

(DBS) and 17 small insertion and deletion (ID) signatures[62]. We extended our analysis of SBS4 to all 77 mutational signatures in PCAWG data and to SBS signatures in TCGA data. We did not investigate mutational signatures in AACR GENIE data due to smaller mutation numbers resulting from the limited genome coverage of the MSK-IMPACT panel. Like our analysis of SBS4, we examined both the proportion of signature-positive tumours as well as relative mutation activity. Previous studies of mutational signatures describe the correlations between age and

signature-attributed mutations but ignore the other aspects of signature detection and relative activity. By comparing signature detection rates, we identified mutational processes that are more likely to be active in younger vs. older patients and vice versa. By analysing signature-attributed mutations as a proportion of total mutations per tumour, we derived information about that signature's contribution to the overall mutational spectrum. For example, SBS1 is well-known as being 'clock-like' and its number of attributed mutations increase with age[60,62]. However, because

**Fig. 2 Age-associated mutational signatures suggest differences in underlying mutational processes. A** In TCGA lung adenocarcinoma adenocarcinoma (LUAD; $n = 497$ biologically independent samples), age (top plots) and PGA (bottom plots) are associated with the tobacco-associated signature SBS4 (left; two-sided Wilcoxon rank sum test). Yellow and blue dots indicate SBS4− and SBS4+ tumours, respectively. The absolute number of SBS4-attributed mutations is also associated with age and PGA (middle; Spearman correlation). The relative proportion of SBS4-attributed mutations is negatively associated with age and has no significant relationship with PGA (right; Spearman correlation). **B** The negative association between PGA and age in LUAD remains significant in both SBS4+ (blue) and SBS4− (yellow) groups (Spearman correlation). **C** Summary of associations between age and the proportion of signature-positive tumours, where dot size shows the marginal log odds from logistic regression and background shading show adjusted multivariate $p$ values. PCAWG data is on left and TCGA on right. **D** Similarly, the summary of associations between age and relative signature activity, with dot size showing Spearman correlations and background indicating adjusted linear regression $p$ values. **E** Comparison of PCAWG and TCGA signature detection frequency. Filled in and open circles indicate comparisons where the differences are statistically significant (proportion test FDR-adjusted $p < 0.05$) and not, respectively. Proposed SBS signature aetiologies are as indicated. Proposed DBS and ID aetiologies are: DBS1: UV, DBS2: tobacco, DBS5: platinum chemotherapy, DBS7: defective MMR, ID1: slippage during DNA replication, ID2: slippage during DNA replication, ID3: tobacco, ID6: defective homologous recombination, ID8: non-homologous end joining, ID13: ultraviolet radiation. Tukey boxplots are shown with the box indicating quartiles and the whiskers drawn at the lowest and highest points within 1.5 interquartile range of the lower and upper quartiles, respectively. Source data are provided as a Source Data file.

SBS1 is detected almost universally, it is equally likely to occur in tumours of younger vs. older patients; when analysed as a proportion of total mutations, we found that the proportion of SBS1 mutations did not change with age, suggesting that its relative activity is stable with age (Spearman's correlation $p > 0.1$).

Across all 2562 PCAWG tumours, we identified twelve mutational signatures with age-associated detection frequency (Fig. 2C, *left*) and ten with age-associated relative signature activity (Fig. 2, *left*). For example, tumours arising in older patients were more likely to be SBS3-positive (marginal log odds change = 0.0085, 95% CI = 0.0024–0.015, adjusted LGR $p = 0.075$), but in these SBS-positive tumours, the proportion of SBS3-attributed mutations decreased with age ($\rho = -0.20$, FDR-adjusted LNR $p = 3.2 \times 10^{-3}$, Bonferroni-adjusted LNR $p = 0.013$). SBS3 mutations are thought to be caused by defective homologous recombination-based DNA damage repair. These results imply that while tumours derived from older individuals were more likely to harbour defective DNA damage repair, its relative impact on the burden of SNVs was lower compared with tumours derived of younger individuals. A similar relationship was seen for ID8, associated with defective non-homologous DNA end-joining (marginal log odds change = 0.024, 95% CI = 0.020–0.028, FDR-adjusted LGR $p = 3.4 \times 10^{-3}$, Bonferroni-adjusted LNR $p = 0.021$; $\rho = -0.099$, FDR-adjusted LNR $p = 3.7 \times 10^{-5}$, Bonferroni-adjusted LNR $p = 3.7 \times 10^{-5}$) and ID1, associated with slippage during DNA replication (marginal log odds change = 0.013, 95% CI = 0.0059–0.020, FDR-adjusted LGR $p = 0.018$; $\rho = -0.059$, FDR-adjusted LNR $p = 0.048$). Like our results for SBS4, we identified associations between age and other tobacco-related signatures DBS2 and ID3. Conversely, tumours arising in older individuals were less likely to exhibit defective base excision repair (SBS36). All mutational signatures findings are in Supplementary Data 3.

These pan-cancer differences persisted across individual tumour-types. We identified 23 age-associated signatures across eleven tumour-types, including six significant signatures in melanoma. In this tumour-type, tumours arising in older patients were preferentially SBS2-positive (marginal log odds change = 0.051, 95% CI = 0.013–0.095, adjusted LGR $p = 0.029$, Fig. 2C), attributed to APOBEC cytidine deaminase activity[63]. Melanomas arising in younger patients were more likely to be positive for signatures related to UV damage (SBS 7a, b, d, Fig. 2C, Supplementary Data 3). The proportion of mutations attributed to UV damage was also higher in younger patients (DBS1, $\rho = -0.29$, FDR-adjusted LNR $p = 0.019$, Fig. 2D), while the proportion of mutations attributed to slippage during DNA replication was higher in older patients (ID1, $\rho = 0.27$, FDR-adjusted LNR $p = 0.019$, Fig. 2D). These results suggest that

melanomas in younger patients more frequently involve UV exposure and damage, while melanomas in older patients were more influenced by endogenous sources of mutation.

Leveraging data describing SBS signatures in TCGA data, we repeated this analysis to identify age-associations in signatures derived from whole exome sequencing (WXS) data. Across pan-TCGA tumours, we detected five signatures that occurred more frequently in older individuals, and three that occurred more frequently in younger individuals (Fig. 2C). We also identified five signatures with higher relative activity in younger patients (Fig. 2D).

There was moderate agreement between TCGA and PCAWG findings: while the results in one dataset never contradicted those of the other, some signatures were associated with age exclusively in either TCGA or PCAWG data. Other signatures, such as SBS1 and SBS5 were associated with age in detection and relative activity across a range of tumour-types in either dataset. There was complete agreement in only some signature like SBS2 and SBS4. We hypothesised that this was due to differences in signature detection rates between WXS and whole genome sequencing (WGS) data and compared how frequently each signature was detected across all samples (Fig. 2E). Signatures with high agreement between datasets had similar detection rates, as observed for SBS2 (detection difference = 1.5%) and SBS4 (detection difference = 1.1%). Signatures where findings did not replicate had vastly different detection rates, as was seen for SBS1 (detection difference = 7.2%) and SBS5 (detection difference = 10%). We further examined this by comparing signatures data from non-PCAWG WGS and non-TCGA WXS data. Differences in signature detection rates between PCAWG and TCGA data were reflected in non-PCAWG WGS and non-TCGA WXS data (Supplementary Fig. 3). We also looked specifically at identified age-associations and found high agreement in data generated by the same sequencing strategy (Supplementary Fig. 2). These findings suggest high confidence in age-associations detected in both WGS and WXS data, and that additional study is needed in independent WXS and WGS data to validate TCGA- and PCAWG-specific findings.

**CNA differences associated with transcriptomic changes.** Global mutation characteristics such as genome instability are features of later stages in a tumour's evolutionary history. In contrast, the early stages are often driven by chromosome- or gene-specific events such as loss of specific chromosomes[55]. CNAs usually affect broad genomic segments containing many genes, but not all these genes confer a selective advantage; algorithms such as GISTIC[64] identify targeted oncogenes and tumour suppressors and have been used to describe catalogues of CNA drivers[65]. We therefore narrowed our focus to 87 known CNA

cancer driver genes as described by COSMIC[66]. We applied our statistical framework to identify putative age-associated copy number driver gains and losses using univariate logistic regression, and those passing a FDR threshold of 10% were modeled using multivariable logistic regression to account for confounding factors. We further used Pearson's $X^2$ tests to evaluate all driver CNAs as an orthogonal measure to minimise false positive hits: we took only results that pass the two stacked 10% FDR thresholds from our statistical framework and the 10% FDR threshold on Chi-squared $p$ values to be significant. We applied these analyses to PCAWG, TCGA and AACR GENIE datasets separately to characterise pan-cancer and tumour-type-specific associations. We assessed our statistical approach with $p$ value Q–Q plots as presented in Supplementary Fig. 4.

In pan-cancer analysis of TCGA data, we identified 20 driver genes that were more frequently lost (Fig. 3A) and eight driver genes that were more frequently gained in in tumours of older individuals (Fig. 3B). Age-associated loss of FANCA (marginal log odds change = 0.015, 95% CI = 0.012–0.018, FDR-adjusted MLR $p = 3.2 \times 10^{-9}$, Bonferroni-adjusted MLR $p = 5.9 \times 10^{-9}$) was also statistically significant in AACR GENIE data (marginal log odds change = 0.051, 95% CI = 0.036–0.066, FDR-adjusted MLR $p = 0.011$, Bonferroni-adjusted MLR $p = 0.011$). Other age-associated loss and gain events were statistically significant in one dataset and corroborated by similar effects in at least one other (Supplementary Data 4–5). There were also age-associated CNAs in specific tumour-types: we detected age-associated gains with evidence in at least two datasets in five tumour-types (Fig. 3B), and losses in six tumour-types (Fig. 3A), most notably in ovarian cancer (Supplementary Data 5). Most of these associations were positive, indicating these CNA drivers were more likely to occur in tumours of older patients.

We next asked whether age-associated CNA drivers might lead to downstream transcriptomic changes by investigating TCGA tumour-matched mRNA abundance data. We used linear models with age, copy number status and their interaction as predictors. These terms tell us when the CNA event itself is significantly associated with mRNA abundance (Fig. 3C top), when mRNA differs by age (Fig. 3C middle) and when the effect of the CNA on mRNA depends on age (Fig. 3C bottom). We adjusted for tumour purity (as estimated by study pathologists) in all mRNA analyses. Of 43 age-associated CNAs with mRNA data, we found 17 were significantly associated with changes in mRNA abundance (Fig. 3C top, Supplementary Data 6). Intriguingly, CDKN2A loss in sarcoma was not itself significantly associated with a change in mRNA, but did significantly interact with age (Fig. 3C bottom). There was a greater decrease in CDKN2A mRNA in sarcomas derived of older than younger individuals (adjusted mRNA-CNA-age $p = 0.024$, Fig. 3E).

To investigate potential clinical significance of these age-associated CNAs, we performed survival analysis to identify prognostic events. We used Cox Proportional-Hazards (Cox PH) models with 5-year overall survival as the end point. Like our mRNA models, we used predictors including copy number status, age and their interaction. In glioblastoma, age itself is a known prognostic feature with older patients having poorer outcome (HR = 2.1, 95% CI = 1.7–2.6, Wald $p = 1.4 \times 10^{-13}$). We found that loss of SUFU was not prognostic, but incorporating age revealed that younger individuals with no SUFU loss had the best outcome (HR = 0.42, 95% CI = 0.30–2.3, adjusted Wald $p = 5.5 \times 10^{-6}$). Also, SUFU loss stratified younger individuals into two groups with distinct trajectories, but had no such prognostic value in glioblastomas of older individuals (Fig. 3E). We repeated these mRNA and survival analyses for all TCGA tumour-types with age-associated CNAs and present all results in Supplementary Data 6.

**SNVs differences associated with functional changes.** Finally, we investigated gene-level SNVs for age-associations. In PCAWG analysis, we used a predefined set of genomic driver and mitochondrial genes[67]. In TCGA analysis, we focused on a set of 679 COSMIC driver genes[66] and applied a recurrence threshold to filter out genes mutated in <1% of tumours. We used AACR GENIE data generated using MSK-IMPACT targeted sequencing of up to 468 cancer genes, and filtered with a recurrence threshold of 1%. We included SNV density in our multivariate models in addition to other confounding factors.

In pan-cancer analysis, we identified 102 age-associated genes in TCGA, nine age-associated genes in AACR GENIE, and one in PCAWG data (Supplementary Data 7). CREBBP-frequency was associated with age in both TCGA (marginal log odds change = 0.030, 95% CI = 0.024–0.040, FDR-adjusted LGR $p = 0.049$) and PCAWG (marginal log odds change = 0.027, 95% CI = 0.0089–0.047, FDR-adjusted LGR $p = 8.7 \times 10^{-3}$, Fig. 4A, Supplementary Data 7). In AACR GENIE, the positive association between CREBBP-status and age was not significant after multiple testing correction (marginal log odds change = 0.011, 95% CI = 0.0047–0.022, FDR-adjusted $p > 0.1$). KDM6A and RBM10 were more likely to have SNV in tumours derived of older patients in TCGA and AACR GENIE but were not recurrently mutated in PCAWG data and not analysed in that dataset (Fig. 4A, Supplementary Data 7). 35 other genes found to be associated with age in TCGA data showed similar effect sizes in either PACWG or AACR GENIE data, but without reaching statistical significance.

There were also tumour-type specific age-associations in SNV frequency in TCGA, PCAWG and AACR GENIE. We identified three tumour-types with consistent and significant age-associated SNVs in at least two datasets, and five tumour-types with associations that were significant in one dataset and showed the same effect in one other (Fig. 4B, Supplementary Data 7). SNVs in FOXA1 occurred more frequently in breast and prostate tumours derived of older individuals. SPOP was also positively associated with age in prostate tumours (PCAWG adjusted LGR $p = 0.099$, AACR GENIE adjusted LGR $p = 0.03$). In melanoma, SNVs in NF1 were more frequent in tumours of older individuals, while BRAF SNVs were more frequent in tumours of younger individuals. We also confirmed known associations between age and mutations in tumour suppressors IDH1- and ATRX- in both high grade glioblastoma and lower grade gliomas, IHD1- and ATRX- were more frequent in tumours derived of younger individuals (Fig. 4C, Supplementary Data 7). Other age-associated SNVs included positive associations in cervical and head and neck cancer, and negative associations in colorectal cancer (Supplementary Data 7).

Like the nuclear genome, the mitochondrial genome is frequently mutated in cancer[68]. We leveraged mitochondrial SNV (mtSNV) data from PCAWG WGS and identified age associated mtSNVs in pan-cancer analysis and in ovarian cancer (Fig. 4D, Supplementary Data 7). All significant age-associations were mtSNVs that occurred more frequently in tumours of older patients, even after controlling for the number of mitochondria copies in each tumour. Implicated mitochondrial regions included MT-CYB, which encodes a cytochrome b and the D-loop, a noncoding region that controls replication and transcription[69,70] (Fig. 4E). We also examined whether the number of tumour mitochondria copies was associated with age by investigating the foldchange compared with normal mitochondria copies. There was indeed a significant association between mitochondria copy increase and age in pan-cancer analysis and three tumour-types (Fig. 4F, Supplementary Data 7). In these tumour contexts, tumours of older patients gained more mitochondria copies than tumours of younger patients.

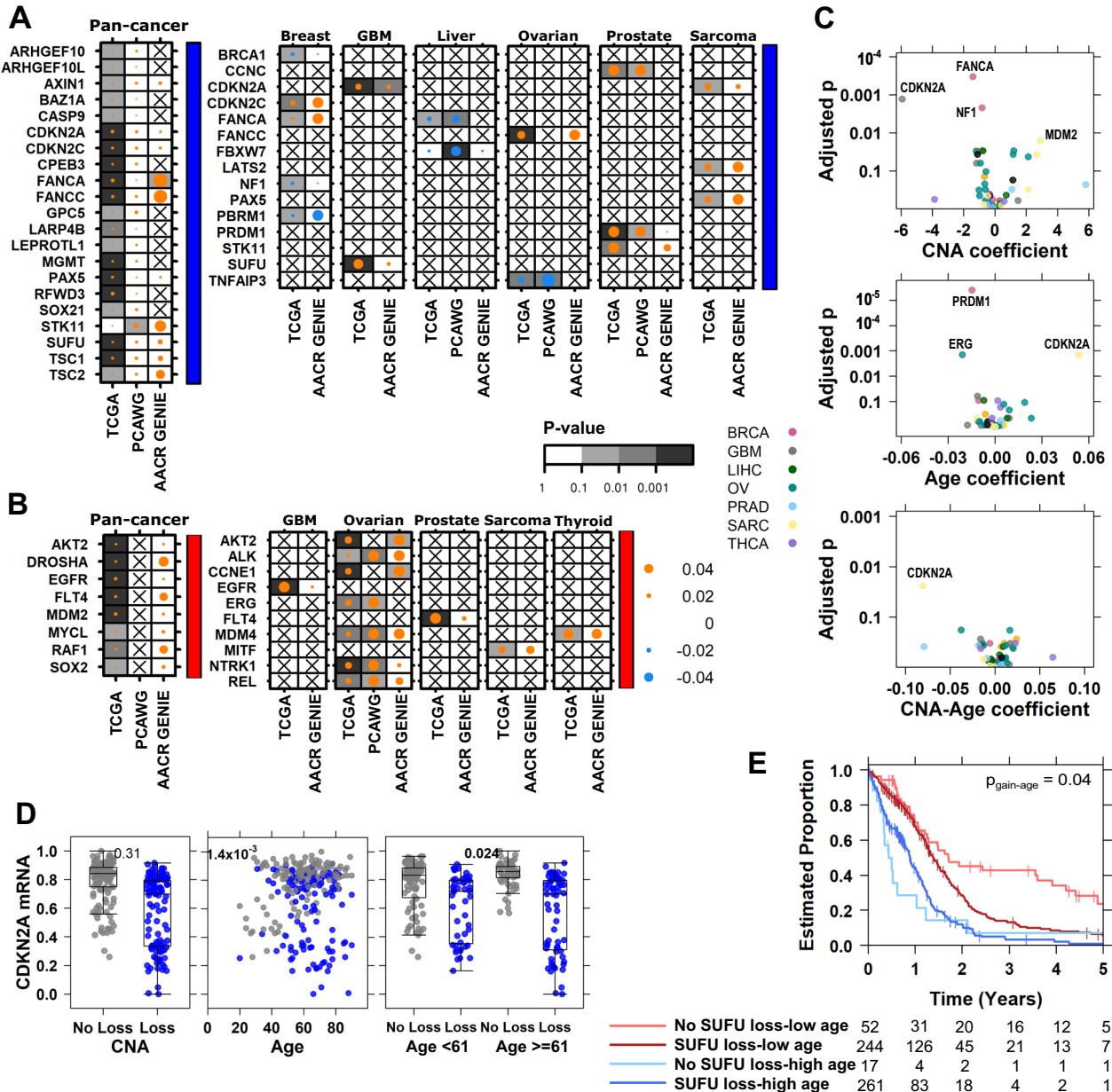

**Fig. 3 Age-associations in copy number drivers are associated with functional changes in mRNA and survival.** Summary of all detected age-associated CNAs for driver (**A**) losses (indicated by blue covariate bar) and (**B**) gains (red covariate bar) across three datasets. Dot size shows the magnitude of the association as the difference in proportion and the background shading shows adjusted multivariate $p$ values. Right covariate indicates copy number gain drivers in red and loss drivers in blue. **C** Age-associated CNAs lead to differential mRNA abundance with respect to the CNA itself (top), age (middle), as well as age-specific effects of the CNA (bottom). **D** In TCGA sarcoma ($n = 255$ biologically independent samples), *CDKN2A* mRNA abundance changes between copy number loss (blue) or no loss (black) and broken down by age for low vs. high age. Tukey boxplots are shown with the box indicating quartiles and the whiskers drawn at the lowest and highest points within 1.5 interquartile range of the lower and upper quartiles, respectively. FDR-adjusted $p$ values from two-sided Wilcoxon rank sum test and Spearman correlations shown. **E** *SUFU* loss in glioblastoma (GBM; $n = 574$ biologically independent samples) interacts with age to further stratify patient prognosis. The adjusted $p$ value for the copy number loss-age interaction term is shown. Source data are provided as a Source Data file.

As with the age-associated CNAs, we evaluated the impact of SNVs on mRNA abundance and survival in TCGA data. We identified significant associations between age-associated SNVs and mRNA abundance for *ATRX* and *IDH1* in lower grade glioma (Supplementary Data 6). Mutations in *ATRX* and *IDH1* were associated with lowered mRNA abundance in both genes. There was also a significant interaction between age and *IDH1*-frequency (adjusted $p = 2.1 \times 10^{-4}$, Fig. 4G) indicating an age-dependent effect on mRNA abundance: mutated *IDH1* was

associated with a greater mRNA decrease in tumours arising in younger patients. Interestingly, this difference was due to a change in baseline *IDH1* mRNA: older patients had higher *IDH1* mRNA abundance than younger, and mutated *IDH1* resulted in equalised mRNA levels. *IDH1* encodes isocitrate dehydrogenase 1, a component of the citric acid cycle: differences in its baseline abundance may be due to differences in metabolism in younger and older brains[71]. In contrast while there was no interaction between age and mutation status on mRNA abundance (adjusted

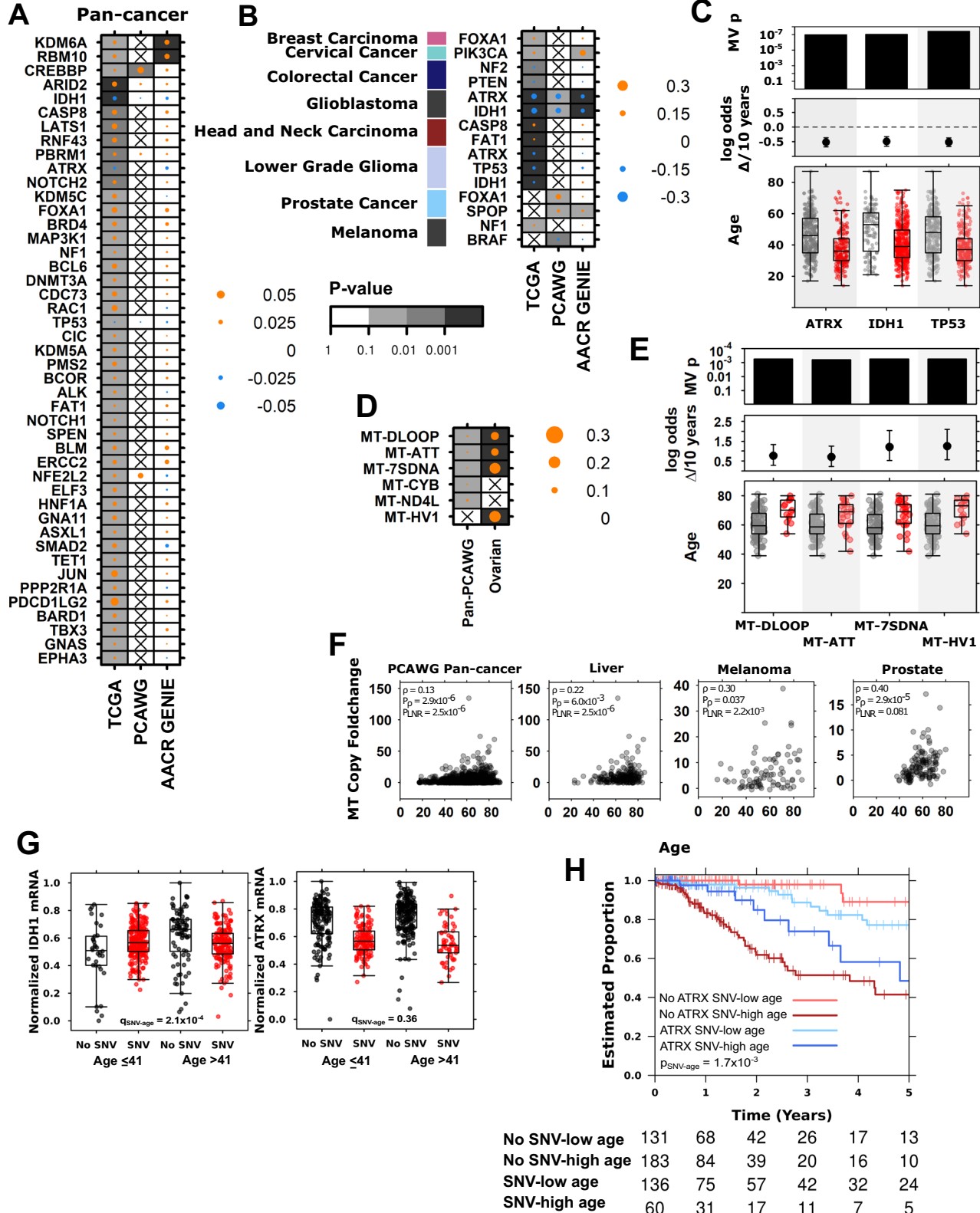

$p = 0.36$, Fig. 4G), *ATRX* and age were synergistically associated with outcome, stratifying lower grade glioma patients into four groups (Fig. 4H).

The lower grade glioma tumour-type is comprised of astrocytoma and oligodendroglioma subtypes. *IDH1* mutation is intrinsically linked to glioma subtype as oligodendroglioma is diagnosed based on the presence of both the 1p/19q co-deletion

and mutation of either *IDH1* or *IDH2*[72]. While our multivariable models adjust for tumour subtype, we investigated age-associations of *ATRX* and *IDH1* SNV mutation frequency in lower grade gliomas in greater detail. We stratified the TCGA lower grade tumours into astrocytoma, oligodendroglioma and oligoastrocytoma subtypes and repeated the SNV, mRNA and survival analyses in each group. We found *TP53* and *ATRX* SNV

**Fig. 4 Age-associations in nuclear and mitochondrial SNVs reveal ATRX as an age-associated prognostic biomarker in lower grade glioma. A** Summary of age- associated nuclear driver SNVs for (**A**) pan-cancer and (**B**) tumour-type specific analyses across three datasets. Dot size shows the magnitude of the association as the difference in proportion and the background shading shows FDR-adjusted multivariate regression (MV) p values. Left covariate in (**B**) indicates relevant tumour-type. **C** TCGA lower grade glioma ($n = 515$ biologically independent samples) age-associations in driver mutation frequency with adjusted multivariate p values, marginal log odds changes for 10-year age increment, and age of tumours compared between those with (red) and without (grey) the mutation. **D** Summary of age-associated mitochondrial SNVs in PCAWG with specific examples from ovarian cancer data ($n = 110$ biologically independent samples) shown in (**E**). **F** Mitochondrial copy number foldchange is also associated with age in four tumour contexts. **G** In TCGA lower grade glioma ($n = 515$ biologically independent samples): mRNA abundance changes for *IDH1* and *ATRX* when the gene is mutated (red) or not (black) compared by median-dichotomised age. Adjusted SNV-age interaction p values are shown. **H** *ATRX* mutation interacts with age to stratify lower grade glioma patient prognosis into four groups. Log-odds p value is shown. Tukey boxplots are shown with the box indicating quartiles and the whiskers drawn at the lowest and highest points within 1.5 interquartile range of the lower and upper quartiles, respectively. For (**C** and **E**): tumours with indicated mutation shown in red, without in grey, and coefficient estimate from linear modeling and 95% confidence intervals shown. Source data are provided as a Source Data file.

mutations occurred more frequently in tumours derived of younger individuals across all three subtypes (Supplementary Fig. 5). *ATRX* SNV mutation frequency was associated with decreased mRNA abundance in all three subtypes, and we also found a significant age-*ATRX* interaction in astrocytoma, where *ATRX*- was associated with a greater decreased in mRNA abundance in tumours of older individuals than younger (interaction $p = 0.016$, Supplementary Fig. 5). *IDH1* SNV mutation frequency was negatively associated with age in astrocytoma (Wilcox $p = 1.3 \times 10^{-8}$) and oligoastrocytoma (Wilcox $p = 0.041$), but not oligodendroglioma (Wilcox $p = 0.19$), and we found significant age-*IDH1* interactions in mRNA abundance analysis for oligodendroglioma (interaction $p = 0.038$) and oligoastrocytoma (interaction $p = 6.2 \times 10^{-3}$). Finally, we found significant age-dependent associations of *ATRX* SNV status in astrocytoma but not oligodendroglioma or oligoastrocytoma: ATRX SNVs were associated with improved survival in older patients, but with worse survival in younger patients.

## Discussion

Despite modest statistical power, suboptimal study designs and limited clinical annotation, we identified myriad age-associated differences in cancer genomes. Age-associated genomic features occur at the pan-cancer level and across almost all individual tumour-types. Combined with similar reports of sex- and ancestry-associated differences in cancer genomes[53,54], these data reveal a set of host influences on the mutational characteristics of tumours. Indeed, a study by Chatsirisupachai et al. describes corroborating evidence of age-associated differences in the genome and transcriptome, as well as age-specific differences in methylation and gene expression control[73]. Together, we find that characteristics of the tumour host appears to influence all aspects of the tumour molecular profile and that some of these lead to age-specific transcriptomic and clinical impacts. We note that most of the age-associated findings in this study survive Bonferroni-adjustment as well as FDR-adjustment which is standard in our field.

The mechanisms for these genomic associations are largely unknown. Our data suggest some endogenous or exogenous mutational processes preferentially occur in individuals of different age groups. Some of these mutational processes are related to aging-associated phenomena such as declining DNA damage repair[74,75], somatic mosaicism and the accumulation of mutations over time[60,76,77]. However, other processes related to immune surveillance, evolutionary selection, disease aetiology and epigenetics are also likely involved[78–80]. Pathogenic germline variants such as those in *BRCA1/2* or *TP53* also lead to earlier presentation of cancer. While our results remain unchanged on removing tumours with detected known pathogenic variants[81], it

is likely there remains hereditary confounders that we have not accounted for.

In addition to such biological factors, lifestyle and socioeconomic considerations like diet[82] and microbiome composition[83] can continuously shape tumour evolution from its earlier steps. Many of these factors are deeply linked to not only an individual's age, but other fundamental characteristics over which we have limited control, such as ancestry or sex. For example, we found that tobacco exposure is closely linked to the negative correlation between age and PGA. It is possible that tobacco exposure leads to earlier presentation of mutation-dense lung cancers. However, it is also likely that there are other variables and interactions that influence the relationship between age and mutation density. Moreover cohort effects, where individuals born in one time period experience different risk exposures from those born in another, can greatly influence the somatic profile of tumours. Our analyses do not consider such cohort effects, and some described age-associations may instead be attributed to differences across time. A tumour's mutational history therefore reflects a complex interplay of biological, lifestyle and healthcare factors, and we have little understanding of how these diverse processes interact to produce molecular phenotypes.

Most tumour-types in our study showed some association between age and genomic alterations. Prostate cancer showed persistent associations in all three datasets across multiple measures of genomic alterations. These were all positive age-associations where mutation density, driver CNA frequency, and SNV frequency of *SPOP* and *FOXA1* all increased with age. The strong association of age with alterations in prostate cancer may be related to its typically slow-growth and low mutation burden. Moreover, known exogenous risk factors, including diet and endocrine disrupting chemicals are thought to converge on hormone regulation, which then acts on the prostate[84]. In contrast, tumours types with negative age-associations such as lung and liver cancers have exogenous risk factors that impact cells through multiple indirect and direct mechanisms such as viral infection, mutagenesis and inflammation[85].

The TCGA, PCAWG and AACR GENIE datasets sometimes identified different molecular associations, highlighting the differences between the three datasets. TCGA and AACR GENIE patients were largely North American while PCAWG had a greater international component. While the ages represented in all three datasets were similar (Table 1), the cohorts differ in other host and clinical characteristics. For instance, the representation of ancestry groups is dissimilar, with many tumour-types differing vastly in ancestry proportions (Supplementary Data 1). Furthermore, differences in sequencing targets also contributed to variation in our results, most conspicuously in the detection rates of some mutational signatures. We customised our analyses to take advantage of the contrasting strengths of each dataset: WGS in PCAWG allowed us to interrogate a greater breadth of

mutation types, while the larger sample size and clinical annotation of TCGA data improved statistical power and controls for confounders. The focused panel used to generate AACR GENIE data facilitated the validation of age-associations in cancer genes. Also, while we were able to identify more age-associations in TCGA data, many of these findings were reflected in PCAWG and AACR GENIE data by similar effects that did not reach our statistical significance threshold. More sequencing data reflecting greater and more balanced diversity is needed to distinguish those age-associations that are intrinsic to differences in biology, and those that are tied to differences in lifestyle and geography.

Our findings have wide-reaching implications for both basic and translational cancer research. Since cancer host characteristics like age, ancestry and sex widely shape the somatic cancer landscape, we cannot consider discovery genomics complete they are explicitly considered. Elderly individuals are underrepresented in cancer sequencing studies and clinical trials[39,86,87]: better inclusion is needed to identify somatic changes specific to older individuals and to leverage these changes to improve clinical care. In our analysis, we found that some age-associated genomic differences associate with transcriptional and clinical changes, but many do not—identifying the functional consequences and mechanisms of these will be a long-term challenge. Finally, these epidemiological factors should be considered and controlled for in personalised therapy strategies. Indeed, every type of analysis from driver-discovery to biomarker-development should explicitly test for and model the powerful influence of patient biology and behaviour on tumour evolution.

## Methods

**Data preprocessing**. For TCGA mRNA abundance, Illumina HiSeq rnaseqv2 level 3 RSEM normalised profiles were converted to $\log_2$ scale. Genes with >75% of tumours having zero reads were removed from the respective dataset. TCGA GISTIC 2.0[64] level 4 data was used for somatic copy-number analysis. mRNA abundance data. Mutational profiles were based on TCGA-reported MutSig v2.0 calls. Genetic ancestry imputed by Carrot-Zhang[88] et al. was incorporated.

We used TCGA data describing 10,212 distinct TCGA tumour samples across 23 tumour-types and 2562 distinct PCAWG samples across 29 tumour-types. Tumour-types with no age annotation were excluded from analysis. Age is treated as a continuous variable for both TCGA and PCAWG analyses. We matched tumour-types between datasets as described in Supplementary Data 1. Source data are provided with this paper.

AACR GENIE-MSK data was generated by the custom hybridisation-based capture panel MSK-IMPACT, which includes up to 410 genes. Preprocessed gene-wise SNV and CNA calls were obtained through the GDC Data Portal. We filtered 6841 metastatic tumours and matched on tumour-types included in the TCGA and PCAWG datasets (Supplementary Data 1) for better comparison across datasets. The final pan-cancer AACR GENIE cohort consisted of 7259 tumours across 35 tumour-types.

**General statistical framework**. For each genomic feature of interest, we used univariate tests first followed by FDR adjustment to identify putative age-associations of interest ($q < 0.1$). We used two-sided non-parametric univariate tests to minimise assumptions on the data. For putative age-associations, we then follow up the univariate analysis with multivariate modeling to account for potential confounders using bespoke models for each tumour-type.

Model variables for each tumour context are presented in Supplementary Data 1 and were included based on availability of data (<15% missing), sufficient variability (at least two levels) and collinearity (as assessed by variance inflation factor). Discrete data was modeled using logistic regression (LGR). Continuous data was first transformed using the Box-Cox family and modeled using LNR. The Box-Cox family of transformations is a formalised method to select a power transformation to better approximate a normal-like distribution and stabilise variance. We used the Yeo-Johnson extension to the Box-Cox transformation that allows for zeros and negative values[89].

FDR adjustment was performed for $p$ values for the age variable significance estimate and an FDR threshold of 10% was used to determine statistical significance. Statistically significant findings must therefore pass two rounds of FDR-adjustment: one at the univariate stage and the second at the multivariate stage. More detail is provided for each analysis below. A summary of all results is presented in Supplementary Data 1 and the subset of all significant results is in Supplementary Data 2. We present 95% confidence intervals for all tests.

**Mutation density**. Performed for both TCGA, PCAWG and AACR GENIE data. Overall mutation prevalence per patient was calculated as the sum of SNVs across all genes on the autosomes and scaled to mutations/Mbp. Coding mutation prevalence only considers the coding regions of the genome, and noncoding prevalence only considers the noncoding regions. TCGA mutation density reflects coding mutation prevalence. AACR GENIE mutation density reflects the targeted sequences of the 341-, 410- or 468-gene MSK-IMPACT panel used for each sample. Mutation density was compared age using Spearman correlation for both pan-cancer and tumour-type specific analysis. Comparisons with univariate $q$ values meeting an FDR threshold of 10% were further analysed using LNR to adjust for tumour subtype-specific variables. Mutation density analysis was performed separately for each mutation context, with pan-cancer and tumour subtype $p$ values adjusted together. Full mutation density results are in Supplementary Data 3.

Per year increase in SNV density was estimated by combining TCGA, PCAWG and AACR GENIE data: for tumour-types with evidence of age-associated SNV density in at least two datasets, we merged those datasets with evidence of age-associations. We fit a LNR model with formula $SNV\ density \sim age + project$ and took the coefficient estimate for age as the per year increase in SNV density value.

**Genome instability**. Performed for TCGA, PCAWG and AACR GENIE data. Genome instability was calculated as the percentage of the genome affected by copy number alterations. The number of base pairs for each CNA segment was summed to obtain a total number of base pairs altered per patient. The total number of base pairs was divided by the number of assayed bases to obtain the percentage of the genome altered (PGA). Genome instability was compared using Spearman correlation for both pan-cancer and tumour-type specific analysis. Comparisons with univariate $q$ values meeting an FDR threshold of 10% were further analysed using LNR to adjust for tumour subtype-specific variables. Genome instability analysis was performed separately for each mutation context, with pan-cancer and tumour subtype $p$ values adjusted together. Full mutation density results are in Supplementary Data 3. Per year increase in PGA was estimated similarly to the estimation for SNV density: we fit a LNR model with formula $PGA \sim age + project$ and took the coefficient estimate for age as the per year increase in PGA. We provide this estimate for tumour-types with evidence of age-associated PGA in at least two datasets.

**Clonal structure and mutation timing analysis**. Performed for PCAGW data only. Subclonal structure data was binarized from number of subclonal clusters per tumour to monoclonal (one cluster) or polyclonal (more than one cluster). Putative age-associations were identified using univariate logistic regression and putative associations were further analysed using multivariate logistic regression. A multivariate $q$ value threshold of 0.1 was used to determine statistically significant age-associated clonal structure.

Mutation timing data classified SNVs, indels and SVs into clonal (truncal) or subclonal groups. The proportion of truncal variants was calculated for each mutation type ($\frac{Number\ truncal\ SNVs}{total\ SNVs}$, etc.) to obtain proportions of truncal SNVs, indels and SVs for each tumour. These proportions were compared using Spearman correlation. Univariate $p$ values were FDR adjusted to identify putatively age-associated mutation timing. Linear regression (LNR) was used to adjust for confounding factors and a multivariate $q$ value threshold of 0.1 was used to determine statistically significant age-a mutation timing. The mutation timing analysis was performed separately for SNVs, indels and SVs. All results for clonal structure and mutation timing analyses are in Supplementary Data 3.

**Mutational signatures analysis**. Performed for TCGA and PCAWG data. For each signature, we compared the proportion of tumours with any mutations attributed to the signatures ("signature-positive") using logistic regression to identify univariate age-associations. Signatures with putative age-associations were further analysed using multivariable logistic regression.

We also compared relative signature activity using the proportions of mutations attributed to each signature. The numbers of mutations per signature were divided by total number of mutations for each tumour to obtain the proportion of mutations attributed to the signature. Spearman correlation was used. Putative age-associated signatures were further analysed using multivariable LNR after Box-cox adjustment.

Signatures that were not detected in a tumour subtype was omitted from analysis for that tumour subtype. All results for clonal structure and mutation timing analyses are in Supplementary Data 3.

**Genome-spanning CNA analysis**. Performed for TCGA, PCAWG and AACR GENIE data. The copy number profiles for COSMIC driver CNA genes were extracted. Copy number calls were collapsed to ternary (loss, neutral, gain) representation by combining loss groups (mono-allelic and bi-allelic) and gain groups (low and high). Logistic regression was used to identify univariate age-associated CNAs. After identifying candidate pan-cancer univariately significant genes, multivariate logistic regression was used to adjust ternary CNA data for tumour-type-specific variables.

We also used Chi-squared tests to evaluate all driver CNAs in all tumour-types. We tested the association of gains/losses with median dichotomised age. Significant

age-associations must pass the two 10% FDR thresholds from our statistical framework and the 10% threshold on FDR-adjusted Chi-squared $p$ values. Bonferroni adjusted $p$ values are also presented in Supplementary Data 2, 5–6. The genome-spanning analysis was performed separately for losses and gains for each tumour subtype.

**Driver SNV analysis**. Performed for TCGA, PCAWG and AACR GENIE data. We focused on driver events described by the PCAWG consortium[67] and by COSMIC[66]. For TCGA and AACR GENIE, we also applied a 1% recurrence filter. Driver mutation data was binarized to indicate presence or absence of the driver event in each patient. Proportions of mutated genes were compared using univariate logistic regression. A $q$ value threshold of 0.1 was used to select genes for further multivariate analysis using binary logistic regression. SNV density was included in all models. FDR correction was again applied and genes with significant age terms were extracted from the models ($q$ value < 0.1). Driver event analysis was performed separately for pan-cancer analysis and for each tumour subtype. All SNV and driver event analysis results are in Supplementary Data 7.

**mRNA functional analysis**. Performed for TCGA data. Genes in bins altered by age-associated CNAs and SNVs after multivariate adjustment were further investigated to determine functional consequences. Tumour purity was included in all mRNA models. Tumours with available mRNA abundance data were matched to those used in CNA analysis. For each gene affected by an age-associated loss, its mRNA abundance was modeled against age, copy number loss status, an age-copy number loss interaction term and tumour purity. The interaction term was used to identify genes with age-associated mRNA changes. FDR adjusted $p$ values and fold-changes were extracted for visualisation. A $q$ value threshold of 0.1 was used for statistical significance. For genes affected by age-associated gains, the same procedure was applied using copy number gains. mRNA modeling results for age-associated CNAs and SNVs are in Supplementary Data 6–7.

**Survival analysis**. Performed for TCGA data. Genes found to have significant (FDR threshold of 10%) age-associated CNAs and SNVs were also analysed using Cox proportional hazards modelling after checking proportional hazards assumption. Cox proportional hazard regression models incorporating age, CNA/SNV status, and an age-CNA/SNV group interaction were fit for overall survival after checking the proportional hazards assumption. Age was treated as a continuous variable for modeling, but median dichotomised into 'low age' and 'high age' groups for visualisation. FDR-adjusted interaction $p$ values and $\log_2$ hazard ratios were extracted for visualisation. A q-value threshold of 0.1 was used to identify genes with sex-influenced survival. Survival modeling results for age-associated CNAs and SNVs are in Supplementary Data 6 and 7.

**Statistical analysis & data visualisation software**. All statistical analyses and data visualisation were performed in the R statistical environment (v3.2.1) using the BPG[90] (v5.9.8) and Survival (v2.44-1.1) packages, and with Inkscape (v0.92.3).

**Reporting summary**. Further information on research design is available in the Nature Research Reporting Summary linked to this article.

## Data availability

The Cancer Genome Atlas datasets were downloaded from Broad GDAC Firehose (https://gdac.broadinstitute.org/), release 2016-01-28. Open access TCGA data was used in this study.

Pan-cancer Analysis of Whole Genomes whole genome sequencing molecular profiles can be downloaded from the PCAWG consortium through the ICGC Data Portal at [https://dcc.icgc.org/releases/PCAWG]: consensus SNV and indels [https://dcc.icgc.org/releases/PCAWG/consensus_snv_indel], consensus copy number data [https://dcc.icgc.org/releases/PCAWG/consensus_cnv], subclonal reconstruction https://dcc.icgc.org/releases/PCAWG/subclonal_reconstruction], and mutational signatures data [https://dcc.icgc.org/releases/PCAWG/mutational_signatures] are available alongside clinical and histology annotation [https://dcc.icgc.org/releases/PCAWG/clinical_and_histology]. PCAWG data is controlled access and administered by dbGaP and ICGC Data Access Compliance Office. Information on accessing the data, including raw read files, can be found at [https://docs.icgc.org/pcawg/data/]. In accordance with the data access policies of the ICGC and TCGA projects, most molecular, clinical and specimen data are in an open tier that do not require access approval. To access potentially identification information, such as germline alleles and underlying sequencing data, researchers will need to apply to the TCGA Data Access Committee (DAC) via dbGaP for access to the TCGA portion of the dataset, and to the ICGC Data Access Compliance Office (DACO) for the ICGC portion. To access somatic single nucleotide variants derived from TCGA donors, researchers will also need to obtain dbGaP authorisation. Researchers may apply for access at [https://docs.icgc.org/download/data-access/].

Controlled access AACR GENIE data was downloaded from AACR Project GENIE for the MSK project[91]. [https://portal.gdc.cancer.gov/projects/GENIE-MSK]. Researchers may apply for access to AACR GENIE from dbGaP under accession number

phs001337.v1.p1 and at [https://gdc.cancer.gov/access-data]. Source data are provided with this paper.

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

## Acknowledgements

The authors thank all the members of the Boutros lab for insightful discussion. We also thank João Pedro de Magalhães, Kasit Chatsirisupachai, Peter Van Loo and the de Magalhães lab for collaborative discussion. This work was supported by the Discovery Frontiers: Advancing Big Data Science in Genomics Research program, which is jointly funded by the Natural Sciences and Engineering Research Council (NSERC) of Canada, the Canadian Institutes of Health Research (CIHR), Genome Canada and the Canada Foundation for Innovation (CFI). P.C.B. was supported by a Terry Fox Research Institute New Investigator Award and a CIHR New Investigator Award. This work was supported by an NSERC Discovery grant and by Canadian Institutes of Health Research, grant #SVB-145586, to PCB. The results described here are in part based upon data generated by the TCGA Research Network: http://cancergenome.nih.gov/. This work was supported by the NIH/NCI under award number P30CA016042 and by an operating grant from the National Cancer Institute Early Detection Research Network (U01CA214194).

## Author contributions

C.H.L. and P.C.B. initiated the project. C.H.L., and S.H. analysed data. P.C.B. supervised research. C.H.L. and P.C.B. wrote the first draft of the paper, which all authors edited and approved.

## Competing interests

The authors declare no competing interests.
