## [Peer review file · Nature Communications]

REVIEWER COMMENTS

Reviewer #1 (Remarks to the Author): expert in cancer genomics, cancer evolution, mutational signatures

Li et al present a pan-can analysis of age related genomic correlates in cancer genomes. The work is ambitious in nature, and certainly addresses a topic of high interest to the scientific community. However the lack of consistency between gene level results from TCGA and PCAWG datasets creates a challenge in drawing definitive conclusions from this work. As suggested below, extending the analysis to additional large scale datasets may help resolve this issue and strengthen the impact of the work. In addition, there are a number of technical questions not clear from the manuscript, such as the level of statistical test inflation (i.e. Q-Q plots) in the gene level CNA analyses. Furthermore, a greater distinction in the signature analysis from other recently published PCAWG work would benefit the manuscript and allow it to present more novelty.

Major comments:

1. In the figure 1 PGA analysis the results are difficult to interpret, given most associations only hold true in one dataset. Other than prostate cancer where a consistent signal is detected. Clearly power may be lacking in PCAWG, particularly for individual cancer types. Given the wealth of other datasets freely available the authors should extend their work to make it adequately powered to answer the question at hand. There are ~4000 whole genomes available from the DRUP trial/Hartwig foundation, see first two links below. In addition, copy number segment data is available for ~30,000 cases sequenced on the MSK-IMPACT panel (which has a genome wide SNP backbone for CN detection), see bottom 2 links.

<https://www.nature.com/articles/s41586-019-1689-y>

<https://www.hartwigmedicalfoundation.nl/en/database/>

<https://sagebionetworks.org/research-projects/aacr-project-genie/>

<https://gdc.cancer.gov/about-gdc/contributed-genomic-data-cancer-research/genie>

2. Is the negative association in lung cancer confounded by smoking status? Never smoker disease, driven by EGFR, etc, particularly in females, can have younger age of onset. The different biology of these tumors may explain the findings, rather than a global age association with PGAs. In tcga (where smoking status is annotated), does the trend hold if never smokers are removed?

3. How does the analysis in figure2 differ from the recently published PCAWG signatures paper:

<https://www.nature.com/articles/s41586-020-1943-3?> In that paper the section titled: "Correlations with age" appears to already present correlations of age versus the same set of signatures, with the same PCAWG data. A more substantially different analysis would be recommended, otherwise the previous work can be cited. Again, linked to point 1 above bringing in additional large datasets (e.g.

Hartwig) beyond the extensively mined TCGA, and PCAWG with a recent dedicated set of papers would be helpful.

4. Can more complete correlation data be presented for copy number CNAs between the two cohorts? In Figure Supp 4 only Ovarian cancer appears to be presented. For the pan-cancer cohort (and perhaps individual cancer types as thumbnail plots), can correlation of gene level $-\log_{10}(p\text{-values})$ (p-value association with age) for tcga and PCAWG be presented, so the reader can assess what level of consistency there is?

5. The number of significantly associated genes in the CNA analysis seems implausibly high – in TCGA 8583 genes associated with loss, and 15497 gain (total ~24k genes). In fact, it seems some genes must be associated with age dependent gain and loss – is this possible? I.e. can a gene be significantly associated with more gains and more losses in older patients. The fig3 results appears contradictory to fig1 data, where the association between age and PGA is modest and only prostate cancer has a reproducible association. Can the authors provide Q-Q plots and inflation values for these sets of tests, to demonstrate there isn't a global inflation in the test statistic? Also, to help verify the large number of associations, a simple simulation may be helpful, where random ages are allocated to

patients and analysis repeated – does this find thousands of significant results?

6. In the SNV analysis, is it just CREBBP that replicates in both TCGA and PCAWG? If so this should be made clearer in the text and main figure.

7. How have germline predisposed patients been dealt with in the analysis, e.g. hereditary BRCA/Lynch syndrome/ Li-Fraumeni syndrome/VHL disease/etc? Were these removed, on account of the bias for earlier age of presentation?

8. In panel 4D the no SNV group appears to have mRNA difference between young and old patients, rather than change in mutated allele. Could the authors provide interpretation of this – is an age dependent reduction in wildtype expression that is driving the pattern?

9. Figure 5 is difficult to understand, and is only mentioned superficially in the text (i.e. in the discussion, not in the results section). A complex figure like this should have a detailed results section, or otherwise a simplified summary style figure developed as a basis for the discussion.

Minor comments:

1. Line 49 – two of the three cancers quoted have screening pathways, and particularly for colorectal the CDC guidelines recommend screening from age 50+. Hence the later stage disease may just reflect a greater rate of presentation outside of screening - can the authors provide additional examples which are not confounded by screening practices.

2. Line 152 – SVNs – should this be SNVs?

Reviewer #2 (Remarks to the Author): expert in statistics

Thank you for the invitation to review the manuscript by Li et al. "Age Influences on the Molecular Presentation of Tumours".

I consider the investigated hypothesis is very interesting, but the conducted data analyses need major substantial revisions. This paper should answer questions like "What is the average increase in SNVs/Mb and small indels/Mb per each year of age at cancer diagnosis, and for Admixed Americans compared to Europeans?"

Statistical framework: Rather than Spearman correlations, I strongly recommend the authors to use multiple linear regression for continuous response variables (SNVs/Mb) and multiple logistic regression for 0/1 outcomes (for example, a particular mutation signature).

FDRs are not appropriate in the context of the conducted investigation. Setting the FDR threshold to 10% means that 10% of the reported findings are false discoveries. The authors should use appropriate methods for multiplicity correction (FWER, simplest approach, just dividing 0.05 by the number of investigated associations in each dataset).

I strongly recommend using standard density measures (SNVs/Mb, small indels/Mb,...) instead of the PGA.

"Bias" has a very specific meaning in statistics, I would recommend the authors substituting "Age biases" by "Age associations" and "Age effects".

The abstract is not very informative, some examples: Specific mutational signatures are associated with age (which ones?), A subset of known cancer driver genes were mutated (which ones?), With clear clinical implications (which ones?).

Please avoid the use of unnecessary abbreviations like ULR, and BLCA, BRCA, CESC, COADREAD,....

The manuscript needs extensive edition by a statistician. For example, alternative formulation for lines 272-272 -> We next asked whether "CNAs associated with age" perturb... Another example on line 275: as predictors (and which was the response variable?)

Line 460: What do you mean with "insufficient variability in ancestry estimation"?

Fig 1: please add regression lines

Fig 1 legend: Mutation density and timing are associated with age at diagnosis

Fig 2: Some kind of interpretation is needed for the mutational signatures

Fig 5 is difficult to understand, I recommend to prepare separated figures for age, gender and ancestry

Reviewer #3 (Remarks to the Author): expert in ageing

Li, Haider and Boutros analyzed cancer genomic datasets from TCGA and PCAWG, generating substantial new information regarding mutational processes that are altered by the age of the individual, including SNV load, CNA, clonality, SNV/indel timing, and various mutational signatures. They also identified specific CNAs, and associated some of these changes with mRNA changes, that changed with age of the patient. Similarly, they identified some age-associated SNVs that associated with mRNA expression. For both these CNAs and SNVs, they demonstrated that these changes interacted with age in prognosis for select cancers (for example, it's interesting that 10q loss is a poor prognosis marker for younger patients but not older ones). Notably, their modeling accounted for confounding variables including sex and genetic ancestry. As they indicate, previous studies (in colon cancers and GBM) have examined age dependent differences in genomic landscapes, but the current study is unique in its breadth (pan-cancer) and in its extensive analyses.

This work is complicated and not "an easy read". A lot of results are presented. Nonetheless, there are some nuggets of important and sometimes intriguing associations with age. For example, lung adenocarcinomas exhibit more CNAs and SNVs in younger patients than older, which is the opposite of that observed for most other tumors. In most cases, the expected increases in mutational burden are observed in cancer in older individuals. It is also interesting that some cancers show increased clonal (truncal) SNVs in older individuals, while the opposite is observed for other cancers. While there is currently no clear explanation for these results, these analyses will serve to stimulate searches for answers. Most importantly, this work should serve as an important resource for understanding how age influences mutational processes in cancer genomics for years to come.

Additional comments:

1) Page 5, line 129. It is indeed interesting that SNVs and CNAs are more abundant in lung cancers from younger individuals, but I'm not sure that I agree with (or understand) how this could relate to "smoking exposure". For one, lung cancers in smokers and never smokers show very similar age distributions (if anything, the cancers occur at a bit older ages for smokers and former smokers). Moreover, SCCs occur almost exclusively in smokers. Their observation requires further explanation (really, speculation).

2) In Fig 3i, it's interesting that only younger patients with loss of a piece of chromosome 10q exhibit worse prognosis. They then indicate that they "performed survival modeling for 5,251 genes on affected by age-biased CNAs in glioblastoma and found 309 1,821 genes showed associations between copy number change and prognosis and 142 genes had significant CNA-age interactions." This

statement requires more clarification. And how divergent are results from those expected based on chance?

3) Page 11, line 358: They write "mutated IDH1 was associated with a greater mRNA decrease in tumours arising in younger patients." I see the opposite, in that those with the SNV exhibit lower expression only in older patients.

4) The authors need to consider cohort effects – the cancers from older individuals came from people born in earlier years than those from younger people, and thus could have experienced differences in lifestyle and exposures.

5) In the Discussion, it would be helpful to highlight some of the more interesting and perhaps unexpected associations with age uncovered in these studies.

Minor:

1) Define ancestry abbreviations in Fig 5. And "density" is misspelled in the figure.

2) The axes need to be better defined, particularly in Suppl Figures like Suppl Fig 2.

3) Supplementary tables are not labeled well, and it was hard to even figure out which Suppl Table was which.

4) For Fig 3A, use of log10 scale would allow better visualization of results.

Signed: James DeGregori

We thank the reviewers for their thoughtful and focused commentary, which has substantially improved this study. The key changes made have been:

- Addition of validation studies in 7,259 AACR GENIE tumours
- Refinement of our CNA and SNV analyses to focus on driver gene sets
- Addition of mitochondrial SNV and copy number analysis in PCAWG WGS data
- Substantial expansions of methodologic and reporting details
- Focused analysis of tobacco on mutation density using both self-reported and imputed mutational signatures tobacco history

Reviewer #1

Li et al present a pan-can analysis of age related genomic correlates in cancer genomes. The work is ambitious in nature, and certainly addresses a topic of high interest to the scientific community. However the lack of consistency between gene level results from TCGA and PCAWG datasets creates a challenge in drawing definitive conclusions from this work. As suggested below, extending the analysis to additional large scale datasets may help resolve this issue and strengthen the impact of the work. In addition, there are a number of technical questions not clear from the manuscript, such as the level of statistical test inflation (i.e. Q-Q plots) in the gene level CNA analyses. Furthermore, a greater distinction in the signature analysis from other recently published PCAWG work would benefit the manuscript and allow it to present more novelty.

We appreciate the reviewer's kind words about the study and have taken to address the technical concerns as outlined below.

1. In the figure 1 PGA analysis the results are difficult to interpret, given most associations only hold true in one dataset. Other than prostate cancer where a consistent signal is detected. Clearly power may be lacking in PCAWG, particularly for individual cancer types. Given the wealth of other datasets freely available the authors should extend their work to make it adequately powered to answer the question at hand. There are ~4000 whole genomes available from the DRUP trial/Hartwig foundation, see first two links below. In addition, copy number segment data is available for ~30,000 cases sequenced on the MSK-IMPACT panel (which has a genome wide SNP backbone for CN detection), see bottom 2 links.

<https://www.nature.com/articles/s41586-019-1689-y>

<https://www.hartwigmedicalfoundation.nl/en/database/>

<https://sagebionetworks.org/research-projects/aacr-project-genie/>

<https://qdc.cancer.gov/about-qdc/contributed-genomic-data-cancer-research/genie>

This is an excellent suggestion, and we exploited the fully freely available AACR GENIE dataset, leveraging 7,259 with mutation and age data available for validation. These have been added systematically through almost every figure of the study. Many results that did not validate in the smaller dataset now replicate in the AACR GENIE dataset. However, because of its panel-sequencing nature there remain a subset of findings (e.g. trinucleotide mutation signatures) that could not be assessed.

2. Is the negative association in lung cancer confounded by smoking status? Never smoker disease, driven by EGFR, etc, particularly in females, can have younger age of onset. The different biology of these tumors may explain the findings, rather than a global age association with PGAs. In tcga (where smoking status is annotated), does the trend hold if never smokers are removed?

We closely investigated the negative association of mutation density with age in lung cancer. We did not find associations between EGFR or other driver mutations with age and mutation density. Nor did we find significant involvement of germline mutations, though this may be due to the small number of known germline risk variants in these data.

We also performed a more in-depth study of tobacco history as described by self-reported smoking history, self-reported pack-years, and by genomically calculated mutational signatures. We did find some intriguing tobacco-related trends, but the negative associations between age and PGA largely remain. We have included additional analysis of smoking history at Lines 156-174 and Figure 1F-G, and of mutational signatures at Lines 188-205 and Figure 2A-B.

3. How does the analysis in figure2 differ from the recently published PCAWG signatures paper: <https://www.nature.com/articles/s41586-020-1943-3>? In that paper the section titled: "Correlations with age" appears to already present correlations of age versus the same set of signatures, with the same PCAWG data. A more substantially different analysis would be recommended, otherwise the previous work can be cited. Again, linked to point 1 above bringing in additional large datasets (e.g. Hartwig) beyond the extensively mined TCGA, and PCAWG with a recent dedicated set of papers would be helpful.

Our mutational signatures analysis examines age at greater detail than in previous studies including the PCAWG signatures (<https://www.nature.com/articles/s41586-020-1943-3>), the TCGA signatures (<https://www.nature.com/articles/nature12477>) and the tobacco smoking mutational signatures (<https://science.sciencemag.org/content/354/6312/618.abstract>) papers. Moreover, these studies describe on the associations of age with overall burden of mutations attributed to the signature, which is confounded by mutation density. Our analyses instead examine signature detection and relative activity and is therefore able to identify differences in relative signature activity. We have included additional detail on how our analyses differ from prior studies at Lines 224-229:

Previous studies of mutational signatures describe the correlations between age and signature-attributed mutations but ignore the other aspects of signature detection and relative activity. For example, SBS1 is well-known as being 'clock-like' and its number of attributed mutations increase with age^{60,62}. However, when analysed as a proportion of total mutations, we find that SBS1 activity is not correlated with age (Spearman's correlation $p > 0.1$).

4. Can more complete correlation data be presented for copy number CNAs between the two cohorts? In Figure Supp 4 only Ovarian cancer appears to be presented. For the pan-cancer cohort (and perhaps individual cancer types as thumbnail plots), can correlation of gene level $-\log_{10}(p\text{-values})$ ($p\text{-value}$ association with age) for tcga and PCAWG be presented, so the reader can assess what level of consistency there is?

We have changed our CNA analysis to focus on known driver gains and losses, which drastically changed our CNA results. We present comparisons between TCGA, PCAWG and AACR-GENIE for all our results when possible throughout the revised paper. In particular, Figure 3 now shows CNA results across all three datasets for better assessment of consistency.

5. The number of significantly associated genes in the CNA analysis seems implausibly high – in TCGA 8583 genes associated with loss, and 15497 gain (total ~24k genes). In fact, it seems some genes must be associated with age dependent gain and loss – is this possible? I.e. can a gene be significantly associated with more gains and more losses in older patients. The fig3 results appears contradictory to fig1 data, where the association between age and PGA is modest and only prostate cancer has a reproducible association. Can the authors provide Q-Q plots and inflation values for these sets of tests, to demonstrate there isn't a global inflation in the test statistic? Also, to help verify the large number of associations, a simple simulation may be helpful, where random ages are allocated to patients and analysis repeated – does this find thousands of significant results?

In our previous submission, we used a segment approach where genes with similar copy number profiles were binned together – this likely led to overestimation of age-associated gains and losses. As mentioned in our response to point #4, we have adjusted our CNA analysis to focus on known cancer drivers. These include 26 driver gains and 61 driver losses. Our CNA results are now more balanced between losses and gains.

6. In the SNV analysis, is it just CREBBP that replicates in both TCGA and PCAWG? If so this should be made clearer in the text and main figure.

CREBBP remains the most consistent hit in our updated driver-focused SNV analysis. We have revised Figure 4 to show this more clearly, and added at Lines 369-374:

“CREBBP-frequency was associated with age in both TCGA (marginal log odds change = 0.030, 95%CI = 0.024 – 0.040, adjusted LGR p = 0.049) and PCAWG (marginal log odds change = 0.027, 95%CI = 0.0089 – 0.047, adjusted LGR p = 8.7×10^{-3} , Figure 4A, Supplementary Table 6). In AACR GENIE, the positive association between CREBBP-status and age was not significant after multiple testing correction (marginal log odds change = 0.011, 95%CI = 0.0047 – 0.022, adjusted p > 0.1).”

7. How have germline predisposed patients been dealt with in the analysis, e.g. hereditary BRCA/Lynch syndrome/ Li-Fraumeni syndrome/VHL disease/etc? Were these removed, on account of the bias for earlier age of presentation?

Huang et al. (<https://www.ncbi.nlm.nih.gov/pmc/articles/PMC5949147/>) analysed pathogenic germline variants in TCGA data and found that 4.1% (428) of pan-cancer tumours harboured pathogenic variants. We removed these samples and saw no changes in our findings. We have added detail in Discussion at Lines 465-467:

“Pathogenic germline variants such as those in *BRCA1/2* or *TP53* also lead to earlier presentation of cancer. While our results remain unchanged on removing tumours with detected known pathogenic variants⁸⁰, it is likely there remains hereditary confounders that we have not accounted for.”

8. In panel 4D the no SNV group appears to have mRNA difference between young and old patients, rather than change in mutated allele. Could the authors provide interpretation of this – is an age dependent reduction in wildtype expression that is driving the pattern?

The reviewer makes an excellent observation. We speculate that the difference in baseline *IDH1* mRNA may be due to aging-related changes in brain metabolism. This may be a feature of the normal brain specifically, or of both normal changes and cancer-related changes. We have added speculation at Lines 416-421:

“Interestingly, this difference results from a change in baseline *IDH1* mRNA: older patients have higher *IDH1* mRNA abundance than younger, and mutated *IDH1* leads to equalised mRNA levels. *IDH1* encodes isocitrate dehydrogenase 1, a component of the citric acid cycle: differences in its baseline abundance may be due to differences in metabolism in younger and older brains⁶⁹.”

9. Figure 5 is difficult to understand, and is only mentioned superficially in the text (i.e. in the discussion, not in the results section). A complex figure like this should have a detailed results section, or otherwise a simplified summary style figure developed as a basis for the discussion.

We have removed Figure 5 from this submission.

1. Line 49 – two of the three cancers quoted have screening pathways, and particularly for colorectal the CDC guidelines recommend screening from age 50+. Hence the later stage disease may just reflect a greater rate of presentation outside of screening - can the authors provide additional examples which are not confounded by screening practices.

We have added pancreatic cancer and soft tissue sarcomas as tumour-types that are more aggressive in young adults. The sentence at Line 49-52 now reads:

“Tumours arising in young adults (< 50 years of age) are often more aggressive: early onset tumours of the prostate²³, breast²⁴, pancreatic^{25,26}, colorectal²⁷, and soft tissue sarcomas²⁸ are diagnosed at higher stages and associated with lower survival.”

And the added references are:

Early onset pancreatic cancer: a controlled trial
(<https://www.ncbi.nlm.nih.gov/pmc/articles/PMC3959307/>)

Early onset pancreatic cancer: Risk factors, presentation and outcome
(<https://www.sciencedirect.com/science/article/abs/pii/S1424390315000290?via%3DiHub>)

Biologic and clinical characteristics of adolescent and young adult cancers: Acute lymphoblastic leukemia, colorectal cancer, breast cancer, melanoma, and sarcoma
(<https://acsjournals.onlinelibrary.wiley.com/doi/full/10.1002/cncr.29871>)

2. *Line 152 – SVNs – should this be SNVs?*

Yes, we have corrected this typo, thank you!

Reviewer #2

Thank you for the invitation to review the manuscript by Li et al. "Age Influences on the Molecular Presentation of Tumours".

I consider the investigated hypothesis is very interesting, but the conducted data analyses need major substantial revisions. This paper should answer questions like "What is the average increase in SNVs/Mb and small indels/Mb per each year of age at cancer diagnosis, and for Admixed Americans compared to Europeans?"

Statistical framework: Rather than Spearman correlations, I strongly recommend the authors to use multiple linear regression for continuous response variables (SNVs/Mb) and multiple logistic regression for 0/1 outcomes (for example, a particular mutation signature).

We very much appreciate the reviewer's comments on our hypothesis and their careful review. We sincerely apologize for the lack of clarity on several aspects of the presentation and analysis noted here. Our statistical analysis starts off with a univariate screen, followed by the multiple linear or multiple logistic regression to control for confounders on only those results that survive that show univariate significance. We use correlations in the initial univariate screen to select candidates of interest in a non-parametric manner. In the multivariate stage of our analysis, we do indeed use multiple linear and logistic regression for continuous and binary outcomes, respectively.

The reviewer also raises the fascinating topic of cross-variable analyses. We agree that these questions warrant investigation, but are not currently feasible. The key limiting factor to these analyses is the lack of representation of non-European ancestry groups in publicly available data. As an example, in many cancer types, there are fewer than 20 non-Caucasian individuals. After adjusting for required covariates, these analyses are not feasible. To be precise, including interaction terms prevents model convergence or generates over-specified models that cannot be fit. We look forward to the emergence of larger multi-ancestric cohorts that will facilitate such studies in the coming years.

FDRs are not appropriate in the context of the conducted investigation. Setting the FDR threshold to 10% means that 10% of the reported findings are false discoveries. The authors should use appropriate methods for multiplicity correction (FWER, simplest approach, just dividing 0.05 by the number of investigated associations in each dataset).

We thank the reviewer for this comment – the use of FDRs in this setting is quite common, we note a set of PCAWG and TCGA papers that have done so below. Given the expectation in this field of reporting of FDRs we have retained them in the text, but also report unadjusted and p-values systematically in our supplementary tables to allow readers to consider alternative versions of the statistical analysis. A few examples of studies using the same strategy on PCAWG & TCGA data:

- Genomic basis for RNA alterations in cancer (<https://www.nature.com/articles/s41586-020-1970-0>)
- Pathway and network analysis of more than 2500 whole cancer genomes (<https://www.nature.com/articles/s41467-020-14367-0>)
- Analyses of non-coding somatic drivers in 2,658 cancer whole genomes (<https://www.nature.com/articles/s41586-020-1965-x>)

- Pathogenic germline variants in 10,389 adult cancers (<https://www.ncbi.nlm.nih.gov/pmc/articles/PMC5949147/>)
- Comprehensive Characterization of Cancer Driver Genes and Mutations ([https://www.cell.com/cell/pdf/S0092-8674\(18\)30237-X.pdf](https://www.cell.com/cell/pdf/S0092-8674(18)30237-X.pdf))

I strongly recommend using standard density measures (SNVs/Mb, small indels/Mb,...) instead of the PGA.

To our knowledge PGA (sometimes also called FGA) is indeed the standard method of considering copy-number density in non-tetraploid tumours. It has been reported by many groups, including our own, in prior studies. A few examples of other groups using it:

- Percent genome alteration and outcomes after radical prostatectomy in African American men. (https://ascopubs.org/doi/abs/10.1200/JCO.2019.37.7_suppl.24)
- The detection and implication of genome instability in cancer (<https://www.ncbi.nlm.nih.gov/pmc/articles/PMC3843371/>)
- Tumor copy number alteration burden is a pan-cancer prognostic factor associated with recurrence and death (<https://www.ncbi.nlm.nih.gov/pmc/articles/PMC6145837/>)

As the reviewer recommends, for density of point mutations we use SNVs/Mbp throughout as this is the established metric for that mutation density feature.

“Bias” has a very specific meaning in statistics, I would recommend the authors substituting “Age biases” by “Age associations” and “Age effects”.

The reviewer is absolutely correct, we have made this change throughout.

The abstract is not very informative, some examples: Specific mutational signatures are associated with age (which ones?), A subset of known cancer driver genes were mutated (which ones?), With clear clinical implications (which ones?).

Given the very limited space available in the abstract, we are unable to describe more than a cursory summary of findings, but we have updated substantially to better provide details. It now reads:

“Cancer is often called a disease of aging. There are numerous ways in which cancer epidemiology and behaviour change with the age of the patient. The molecular bases for these relationships remain largely underexplored. To characterize them, we analyzed age-associations in the nuclear and mitochondrial somatic mutational landscape of 20,033 tumours across 35 tumour-types. Age influences both the number of mutations in a tumour and their evolutionary timing. Specific mutational signatures are associated with age, reflecting differences in exogenous and endogenous oncogenic processes such as a greater influence of tobacco use in the tumours of younger patients, but more activity of DNA damage repair signatures in those of older patients. We find that known cancer driver genes are mutated in age-associated frequencies, and these alter the transcriptome and predict for clinical outcomes. These effects are most striking in brain cancers where alterations like SUFU loss and ATRX mutation are age-dependent prognostic biomarkers. Using three cancer datasets, we show that age shapes the somatic mutational landscape of cancer, with clear clinical implications.”

Please avoid the use of unnecessary abbreviations like ULR, and BLCA, BRCA, CESC, COADREAD,....

These abbreviations are the standard tumour-type abbreviations used by TCGA & PCAWG, and so we have elected to retain them for consistency with the source data on which these analyses were performed. However, we have added reader-friendly labels as well throughout figures and text (eg: PRAD, Prost-AdenoCA, and Prostate Cancer)

The manuscript needs extensive edition by a statistician. For example, alternative formulation for lines 272-272 -> We next asked whether “CNAs associated with age” perturb... Another example on line 275: as predictors (and which was the response variable?)

We greatly appreciate this caution, and our manuscript has now been extensively edited with statistical terminology in mind. The specific line in question (Line 325-326) now reads:

“We next asked whether age-associated CNAs might lead to downstream transcriptomic changes by investigating TCGA tumour-matched mRNA abundance data.”

Line 460: What do you mean with “insufficient variability in ancestry estimation”?

We apologize for the lack of clarity in our writing, this line has been removed from the text and we have added additional material to clarify our analysis approach

Fig 1: please add regression lines

We have added regression lines as suggested.

Fig 1 legend: Mutation density and timing are associated with age at diagnosis

We have changed the Figure title as suggested.

Fig 2: Some kind of interpretation is needed for the mutational signatures

We have included additional interpretation to the main text describing Figure 2, including a specific example breaking down SBS4 to precede the general discussion of mutational signatures

Fig 5 is difficult to understand, I recommend to prepare separated figures for age, gender and ancestry

As recommended by multiple reviewers, we have removed Figure 5 from our submission.

Reviewer #3

Li, Haider and Boutros analyzed cancer genomic datasets from TCGA and PCAWG, generating substantial new information regarding mutational processes that are altered by the age of the individual, including SNV load, CNA, clonality, SNV/indel timing, and various mutational signatures. They also identified specific CNAs, and associated some of these changes with mRNA changes, that changed with age of the patient. Similarly, they identified some age-associated SNVs that associated with mRNA expression. For both these CNAs and SNVs, they demonstrated that these changes interacted with age in prognosis for select cancers (for example, it's interesting that 10q loss is a poor prognosis marker for younger patients but not older ones). Notably, their modeling accounted for confounding variables including sex and genetic ancestry. As they indicate, previous studies (in colon cancers and GBM) have examined age dependent differences in genomic landscapes, but the current study is unique in its breadth (pan-cancer) and in its extensive analyses.

This work is complicated and not “an easy read”. A lot of results are presented. Nonetheless, there are some nuggets of important and sometimes intriguing associations with age. For example, lung adenocarcinomas exhibit more CNAs and SNVs in younger patients than older, which is the opposite of that observed for most other tumors. In most cases, the expected increases in mutational burden are observed in cancer in older individuals. It is also interesting that some cancers show increased clonal (truncal) SNVs in older individuals, while the opposite is observed for other cancers. While there is currently no clear explanation for these results, these analyses will serve to stimulate searches for answers. Most importantly, this work should serve as an important resource for understanding how age influences mutational processes in cancer genomics for years to come.

We thank the reviewer for their careful and kind assessment of our work. We acknowledge that the work is complicated and dense, and with their guidance we have tried to de-clutter the analysis and present our findings more succinctly. We address specific comments below.

1) Page 5, line 129. It is indeed interesting that SNVs and CNAs are more abundant in lung cancers from younger individuals, but I'm not sure that I agree with (or understand) how this could relate to “smoking exposure”. For one, lung cancers in smokers and never smokers show very similar age distributions (if anything, the cancers occur at a bit older ages for smokers and former smokers). Moreover, SCCs occur almost exclusively in smokers. Their observation requires further explanation (really, speculation).

We have added a focused analysis of smoking in lung cancers and how the age-PGA correlation varies by smoking history. We performed this analysis using both self-reported smoking history and tobacco exposure as described by mutational signatures data. As Figure 1G now shows, age and PGA remain negatively correlated in current and recent reformed smokers (≤ 15 years); the correlation is not significant in never and long-term reformed smokers (> 15 years). Tobacco exposure itself is known to be associated with high mutation burden, and earlier age of diagnosis – the latter becomes clearer in Figure 1F when compared by smoking history group. The combination of these factors may explain higher PGA in younger tobacco-exposed lung cancer patients. Of course, smoking is only one factor we considered in this analysis, and the causes of age-associated differences are likely multifactorial and interacting.

We have added more detail on this analysis to our main text at Lines 156-174, 188-205, and to our Discussion at Lines 472-474:

“For example, we found that tobacco exposure is closely linked to the negative correlation between age and PGA. It is possible that tobacco exposure leads to earlier presentation of mutation-dense lung cancers. However, it is also likely that there are other variables and interactions that influence the relationship between age and mutation density.”

2) In Fig 3i, it's interesting that only younger patients with loss of a piece of chromosome 10q exhibit worse prognosis. They then indicate that they “performed survival modeling for 5,251 genes on affected by age-biased CNAs in glioblastoma and found 309 1,821 genes showed associations between copy number change and prognosis and 142 genes had significant CNA-age interactions.” This statement requires more clarification. And how divergent are results from those expected based on chance?

Our previous analysis was based on chromosome segments obtained by binning adjacent genes with similar CNA profiles. It likely overestimated the number of age-associated gains and losses. We have updated our analysis in this submission to focus on known cancer drivers (26 gains and 61 losses), and our list of CNA-age interactions has changed accordingly. We now describe at Line 353-355:

“Genes in these CNAs were associated with altered mRNA abundance in six tumour-types. Five tumour-types also showed that age-associated CNAs can be prognostic and that the prognostic value can also differ based on the age of the individual.”

3) Page 11, line 358: They write “mutated IDH1 was associated with a greater mRNA decrease in tumours arising in younger patients.” I see the opposite, in that those with the SNV exhibit lower expression only in older patients.

Yes, this is a great point. We speculate the difference in baseline IDH1 mRNA may be due to aging-related changes in brain metabolism. This may be a feature of the normal brain specifically, or of both normal changes and cancer-related changes. We have pointed out the dynamics of the mRNA change and added this speculation at Lines 416-421:

“Interestingly, this difference results from a change in baseline *IDH1* mRNA: older patients have higher *IDH1* mRNA abundance than younger, and mutated *IDH1* leads to equalised mRNA levels. *IDH1* encodes isocitrate dehydrogenase 1, a component of the citric acid cycle: differences in its baseline abundance may be due to differences in metabolism in younger and older brains⁶⁹.”

4) The authors need to consider cohort effects – the cancers from older individuals came from people born in earlier years than those from younger people, and thus could have experienced differences in lifestyle and exposures.

This is also an excellent point. While we control for specific exposures, we do not consider cohort effects in this analysis. It is a weakness of our study and we add mention to it in Discussion at Lines 476-279:

“Moreover cohort effects, where individuals born in one time period experience different risk exposures from those born in another, can greatly influence the somatic profile of tumours. Our analyses do not consider such cohort effects, and some described age-associations may instead be attributed to differences across time.”

5) In the Discussion, it would be helpful to highlight some of the more interesting and perhaps unexpected associations with age uncovered in these studies.

We have expanded our Discussion on findings of interest and comparisons between datasets.

Minor:

1) Define ancestry abbreviations in Fig 5. And “density” is misspelled in the figure.

We have removed Figure 5. Though our paper no longer explicitly address ancestry, we define the abbreviations when appropriate in supplementary materials.

2) The axes need to be better defined, particularly in Suppl Figures like Suppl Fig 2.

We have updated the supplementary figures and adjusted the figure labels.

3) Supplementary tables are not labeled well, and it was hard to even figure out which Suppl Table was which.

We have revised the supplementary tables to be clearer and added better description.

4) For Fig 3A, use of log10 scale would allow better visualization of results.

This worked nicely, we have revised Figure 3 as suggested!

REVIEWER COMMENTS

Reviewer #1 (Remarks to the Author):

The revised manuscript includes additional data and revised analysis. However some of the issues have been unaddressed, or explained without sufficient detail, and major issues remain which prevent support for publication. Here is a specific review of the points:

points 1 and 2 - addressed

point 3 - This is not fully addressed.

The rebuttal states that "Our mutational signatures analysis examines age at greater detail than in previous studies including the PCAWG signatures" however no specifics are given on what the greater detail is. From reading the rebuttal, the main difference seems to be prior studies use absolute number of mutations, whereas this study uses proportion. For signature 1a clock like mutations, which correlate with age based on the assumption of accumulating errors during cell division, it's not clear that using proportion actually makes much sense. For example if a tumor bears $n=1000$ signature 1 mutations, and $n=1000$ other, the signature 1 proportion is 0.5. But if this patient now smokes, and has $n=9000$ signature 4 tobacco mutations, the proportion becomes 0.1 and hence the age estimate is now 5-fold less, but with no underlying difference in absolute mutations associated with cell division errors. This analysis make risk confusion and furthermore, it seems technical rather than biological in nature. Overall the data in figure 2 risks confusion, and it's not clear how it's substantially different from prior publications of the exact same data.

point 4&5 - not addressed

The authors state "We have changed our CNA analysis to focus on known driver gains and losses, which drastically changed our CNA results" and "In our previous submission, we used a segment approach where genes with similar copy number profiles were binned together – this likely led to overestimation of age-associated gains and losses"

It's unclear if the methodology is erroneous, the authors themselves suggest it's leading to overestimated results. If the method is wrong, then simply applying it to a smaller set of genes is a flawed approach. While not as many erroneous results are reported, the results are still overestimated. As requested by the reviewer, were QQ plots reviewed to assess for evidence of systematic inflation in the test statistic? No QQ are presented in the manuscript, and no further detail is given, so it appears data in figure 3 still has major issues.

Points 6-9 - are addressed

Reviewer #2 (Remarks to the Author):

Thank you for the invitation to review the updated version of the manuscript by Li et al. "Age Influences on the Molecular Presentation of Tumours".

Unfortunately, I consider the authors did not adequately address my suggestions:
- Despite the title of the article, the following question remains unanswered: "What is the average increase in SNVs/Mb and small indels/Mb per each year of age at cancer diagnosis (overall and for

particular cancer types)?

- Instead of the estimated effect sizes (SNVs/small indels per Mb and year of diagnosis age) from multiple regression models, the authors continue reporting Spearman correlations only.
- As pointed out in my previous review, FDRs are not appropriate in the context of the conducted investigation – even if FDRs are frequently used in other PCAWG and TCGA publications. The FDR threshold of 10% means that, on average, 10% of the reported findings are false, which I consider is not acceptable.
- The authors state that “As the reviewer recommends, for density of point mutations we use SNVs/Mbp throughout”. Unfortunately, I only see Spearman correlations with the corresponding probability values.
- I still consider the abstract is not very informative. The relationship between mutation density and age at diagnosis remains unquantified, the specific mutational signatures/mutated cancer driver genes associated with age are not listed, the clinical implications are not specified. An appropriate multiplicity correction (FWER) would allow the authors to tighten the results.
- Fig 2: Please add the proposed aetiology for mutations with this information: for example APOBEC activity, defective HR DNA / DNA mismatch repair, UV light exposure,...

Reviewer #3 (Remarks to the Author):

The authors fully addressed my previous concerns, and the major points of the manuscript are now easier to follow. The authors also appear to have addressed concerns of the other reviewers (although some of the points on statistical analyses were outside my sphere of expertise).

signed - James DeGregori

We thank the reviewers again for their continued time and comments. In addition to correcting or updating several minor issues as recommended, we have made two major additions to the paper:

- Q-Q plots for the p-value distributions for our CNA results to Supplementary Figures
- Bonferroni-adjusted p-values in supplementary materials

Reviewer #1:

The rebuttal states that "Our mutational signatures analysis examines age at greater detail than in previous studies including the PCAWG signatures" however no specifics are given on what the greater detail is. From reading the rebuttal, the main difference seems to be prior studies use absolute number of mutations, whereas this study uses proportion. For signature 1a clock like mutations, which correlate with age based on the assumption of accumulating errors during cell division, it's not clear that using proportion actually makes much sense. For example if a tumor bears $n=1000$ signature 1 mutations, and $n=1000$ other, the signature 1 proportion is 0.5. But if this patient now smokes, and has $n=9000$ signature 4 tobacco mutations, the proportion becomes 0.1 and hence the age estimate is now 5-fold less, but with no underlying difference in absolute mutations associated with cell division errors. This analysis make risk confusion and furthermore, it seems technical rather than biological in nature. Overall the data in figure 2 risks confusion, and it's not clear how it's substantially different from prior publications of the exact same data.

The reviewer's example illustrates the value in analyzing absolute signature-attributed mutation numbers. These analyses allow us to compare the intensity of mutational processes and have been described by previous groups. We assert that our proportional approach adds information on other aspects of mutational signatures and do have biological meaning. We compare the binary metric of whether a signature is detected or not, to identify signatures that are more likely to affect younger vs. older patients. We pair this with analyses of proportional mutation activity to add context of how active each signature is in a tumour relative to other detected signatures. In the reviewer's example, the patient in both scenarios has 1000 signature 1 mutations, but significantly more smoking mutations in the second scenario. Our analysis would pick up on that greater activity of the smoking signature. In a cohort, higher proportions of smoking signatures in younger patients suggests that smoking plays a greater mutagenic role in their tumours than in the tumours of older patients. To our knowledge, this is the first time mutational signatures data has been analysed in this way with respect to age. We have added Lines 240-251 in the main text to clarify the interpretations of our analyses:

"Previous studies of mutational signatures describe the correlations between age and signature-attributed mutations but ignore the other aspects of signature detection and relative activity. By comparing signature detection rates, we identify mutational processes that are more likely to be active in younger vs older patients and vice versa. By analyzing signature-attributed mutations as a proportion of total mutations per tumour, we can derive information about that signature's contribution to the overall mutational spectrum. For example, SBS1 is well-known as being 'clock-like' and its number of attributed mutations increase with age^{60,62}. However, because SBS1 is detected almost universally, it is equally likely to occur in tumours of younger vs older patients; when analysed as a proportion of total mutations, we find that the proportion

of SBS1 mutations does not change with age, suggesting that its relative activity is stable with age (Spearman's correlation $p > 0.1$)."

point 4&5 - not addressed

The authors state "We have changed our CNA analysis to focus on known driver gains and losses, which drastically changed our CNA results" and "In our previous submission, we used a segment approach where genes with similar copy number profiles were binned together – this likely led to overestimation of age-associated gains and losses"

It's unclear if the methodology is erroneous, the authors themselves suggest it's leading to overestimated results. If the method is wrong, then simply applying it to a smaller set of genes is a flawed approach. While not as many erroneous results are reported, the results are still overestimated. As requested by the reviewer, were QQ plots reviewed to assess for evidence of systematic inflation in the test statistic? No QQ are presented in the manuscript, and no further detail is given, so it appears data in figure 3 still has major issues.

To clarify, our previous assertion, "In our previous submission, we used a segment approach where genes with similar copy number profiles were binned together – this likely led to overestimation of age-associated gains and losses" referred to the large number of genes contained in each CNA bin. Each bin contained tens to hundreds of genes, and since our CNA analysis methods were bin-based, all genes within each bin shared the same statistical result (same p-value, effect-size, etc.). If there were 100 bins, all but one bin containing 200 genes and the hundredth containing 10,000 genes (purely illustrative example), then if the hundredth bin had an age-associated CNA profile, we would see an inflation in the number of significant hits.

Copy number alterations are large, structural events that alter cancer driver genes, but also the genes around them. Because we are most interested in the age-associations of cancer drivers, we revised our analysis to focus on a pre-defined set of CNA drivers and avoid the issue of picking up spurious CNA passengers. The CNA drivers are more evenly distributed over the genome and are more likely to have independent CNA profiles from each other.

We have prepared Q-Q plots as originally requested by the reviewer and apologize for the omission. They are presented in **Supplementary Figure 4** and reproduced below. They show that the p-values of both univariate and multivariate tests largely follow a uniform distribution with a tail, as desired. There are tumour-types where one dataset shows an un-ideal distribution, but the other two show good results: these cases demonstrate the importance of using multiple independent cohorts. Also, some multivariate Q-Q plots have un-ideal distributions (and fewer points), and these factors may be due in part to the filtering performed in the univariate stage. We agree with the reviewer that the Q-Q plots are important in diagnosing the appropriateness of our statistical approach and have included these plots in our submission for transparency so that readers can better assess our results. We attach Supplementary Figure 4 below:

Supplementary Figure 4. Q-Q plots of CNA-age p-values vs. expected uniform distribution.

Comparison of $-\log_{10}(\text{p-value})$ distributions for univariate (UV) and multivariate (MV) CNA-age association tests against expected univariate distributions. Only tumour-types with significant multivariate results shown (as seen in **Figure 3A-C**).

Reviewer #2:

Thank you for the invitation to review the updated version of the manuscript by Li et al. "Age Influences on the Molecular Presentation of Tumours".

Unfortunately, I consider the authors did not adequately address my suggestions:
- Despite the title of the article, the following question remains unanswered: "What is the average increase in SNVs/Mb and small indels/Mb per each year of age at cancer diagnosis (overall and for particular cancer types)?

- Instead of the estimated effect sizes (SNVs/small indels per Mb and year of diagnosis age) from multiple regression models, the authors continue reporting Spearman correlations only.

As requested we have added estimates of the per year increase in SNV density and PGA by merging datasets with evidence of age associations. We used linear regression models with formulae $\text{SNV density} \sim \text{age} + \text{project}$, and $\text{PGA} \sim \text{age} + \text{project}$ where the project term accounts for differences between datasets. These estimates and 95% confidence intervals are accumulated for all cancer types in Table 2. We have added the following passages to the text:

Lines 113-115:

"Using TCGA and PCAWG data, we estimate that SNV density increases at a rate of 0.077 mutations per megabase pair per year (**Table 2, Methods**)."

Lines 115-122:

"We also identified positive associations in 11 TCGA, 14 PCAWG, and six AACR GENIE tumour-types (**Figure 1A**). Of these, nine tumour-types showed consistent results in two of three datasets (**Supplementary Figure 1, Supplementary Table 2**) including prostate cancer (TCGA: $\rho = 0.25$, FDR-adjusted LNR $p = 0.015$, Bonferroni-adjusted LNR $p = 0.13$; PCAWG: $\rho = 0.48$, FDR-adjusted LNR $p = 1.2 \times 10^{-4}$, Bonferroni-adjusted LNR $p = 8.7 \times 10^{-4}$; **Figure 1B**). Estimates for per year increase in mutation density are given in Table 2 for the nine tumour-types with consistent evidence in at least two datasets."

Lines 153-157:

"We found that in pan-cancer analysis, PGA increased with age in PCAWG ($\rho = 0.19$, FDR-adjusted LNR $p = 0.022$, Bonferroni-adjusted LNR $p = 0.068$) and AACR GENIE ($\rho = 0.041$, FDR-adjusted LNR $p = 0.050$, Bonferroni-adjusted LNR $p = 0.16$) (**Figure 1D**) and estimate that PGA increases at 0.010% per year (**Table 2**)."

Lines 170-171:

"Estimates for per year increase in PGA are given in **Table 2** for the five tumour-types with consistent evidence in at least two datasets."

And in Methods at lines 638-642:

“Per year increase in SNV density was estimated by combining TCGA, PCAWG and AACR GENIE data: for tumour-types with evidence of age-associated SNV density in at least two datasets, we merged those datasets with evidence of age-associations. We fit a linear regression model with formula $SNV\ density \sim age + project$ and took the coefficient estimate for age as the per year increase in SNV density value.”

And lines 654-658:

“Per year increase in PGA was estimated similarly to the estimation for SNV density: we fit a linear regression model with formula $PGA \sim age + project$ and took the coefficient estimate for age as the per year increase in PGA. We provide this estimate for tumour-types with evidence of age-associated PGA in at least two datasets.”

- As pointed out in my previous review, FDRs are not appropriate in the context of the conducted investigation – even if FDRs are frequently used in other PCAWG and TCGA publications. The FDR threshold of 10% means that, on average, 10% of the reported findings are false, which I consider is not acceptable.

We respectfully note that literally hundreds of PCAWG and TCGA publications have reported both FDR-adjusted p-values, and it is unclear to us the justification for why FDRs are not appropriate in the context of this study. Indeed a co-submitted manuscript on similar topics has now been accepted at this journal using FDR throughout without issue (<https://www.nature.com/articles/s41467-021-22560-y>), further highlighting our reservations. Nevertheless, given the reviewer’s strong insistence on this topic we now provide Bonferroni-adjusted p-values. Our key findings in SNV density and PGA, smoking- and homologous repair-associated mutational signatures, and prognostic *SUFU* loss and *ATRX* mutation all remain statistically significant with this much more rigorous correction. Further, we have made available all unadjusted and adjusted p-values for full transparency to *Nature Communications* readership, and in case readers wish to calculate FDRs themselves for comparison with other studies in the TCGA and PCAWG literature. We trust that these changes now address the concern.

We add at Lines 92-97:

“We perform multiple testing adjustment at both stages using the Benjamini-Hochberg false discovery rate (FDR) procedure and these adjusted p-values are used throughout. Bonferroni-adjusted p-values provide similar support for our findings. FDR-adjusted p-values are reported unless otherwise noted. Both Benjamini-Hochberg and Bonferroni, as well as unadjusted p-values are provided in supplementary materials.”

And have similarly updated Supplementary Tables 2-6.

- The authors state that “As the reviewer recommends, for density of point mutations we use SNVs/Mbp throughout”. Unfortunately, I only see Spearman correlations with the corresponding probability values.

The original reviewer comment recommended using SNVs/Mbp rather than PGA. Our initial submission already did so, and of course SNVs/Mbp and PGA reflect different aspects of tumour biology (point variants and copy number aberrations, respectively). This remark appears unrelated to that reviewer comment, and appears to be a reiteration of the request above for indicating per-year changes in SNVs/Mbp. Those changes are outlined above.

- I still consider the abstract is not very informative. The relationship between mutation density and age at diagnosis remains unquantified, the specific mutational signatures/mutated cancer driver genes associated with age are not listed, the clinical implications are not specified. An appropriate multiplicity correction (FWER) would allow the authors to tighten the results.

We are a little confused by this comment. The Abstract does indeed directly refer to two specific mutational signatures (smoking and DNA damage repair) and two specific cancer drivers (*SUFU* and *ATRX*) with prognostic implications in brain cancers. We now describe age-associations with mutation density in the manuscript, but not in the abstract as we believe the mutational process and prognostic gene associations are more important given the space constraints of an Abstract. We are happy to take our lead from the Editor on this issue and will gladly update the abstract as they recommend. We reproduce our abstract below:

“Cancer is often called a disease of aging. There are numerous ways in which cancer epidemiology and behaviour change with the age of the patient. The molecular bases for these relationships remain largely underexplored. To characterize them, we analyzed age-associations in the nuclear and mitochondrial somatic mutational landscape of 20,033 tumours across 35 tumour-types. Age influences both the number of mutations in a tumour and their evolutionary timing. Specific mutational signatures are associated with age, reflecting differences in exogenous and endogenous oncogenic processes such as a greater influence of tobacco use in the tumours of younger patients, but more activity of DNA damage repair signatures in those of older patients. We find that known cancer driver genes are mutated in age-associated frequencies, and these alter the transcriptome and predict for clinical outcomes. These effects are most striking in brain cancers where alterations like *SUFU* loss and *ATRX* mutation are age-dependent prognostic biomarkers. Using three cancer datasets, we show that age shapes the somatic mutational landscape of cancer, with clinical implications.”

- Fig 2: Please add the proposed aetiology for mutations with this information: for example APOBEC activity, defective HR DNA / DNA mismatch repair, UV light exposure,...

We have added SBS signature aetiologies in the figure (**Figure 2E**). Due to space constraints, we have added DBS and ID signature aetiologies in the figure legend.

REVIEWER COMMENTS

Reviewer #1 (Remarks to the Author):

Li et al have submitted a further revised manuscript, with additional explanation and analysis added. Of the two points that were unresolved:

- The point on mutational signatures is now at least clearly explained. Personally, I still struggle to understand if the difference between proportion and absolute signature measures is biological or technical in nature. However the authors have answered my question, and so I would leave this now as an editorial matter. As highlighted in previous review, the analysis presented is very similar to prior published work (in a nature group journal), in the same PACAWG samples. If the editors are happy analysing proportion rather than absolute count vs age is significantly different then the matter should be closed.

- The last point (on copy number analysis, points 4 & 5 from the original rebuttal) is more concerning however - as the author acknowledges, the Q-Q plots have some "un-ideal distributions". The $y=x$ line is missing from the plots (and should be added), but from looking at the data the observed p-values are highly inflated compared to what would be expected. This is a concern as it suggests that nearly any gene could come up as a hit in this analysis, which was also an issue from the first rebuttal where thousands of genes were shown to be significantly associated with aging. This is coupled with the concerns of the other reviewer (who is unsure about multiplicity correction). In the latest reply the authors have provided helpful insight into the root cause of the issue, ie that individual SCNA segments span tens or even hundreds/thousands of genes. This is indeed a challenging issue, and in other major studies (eg TCGA) used specialised analysis tools such as GISTIC were used for SCNA gene level analysis (eg see here for a recent example <https://pubmed.ncbi.nlm.nih.gov/32203465/>). I appreciate it's late in the review process to undertake new primary analysis, but if the key findings from figure 3 could be validated using an orthogonal statistical approach that would reassure greatly the readership. For example, if FANCA loss is identified as a significant GISTIC peak in older individuals, but not (or less markedly less strongly) associated in younger individuals, that would show the key findings are valid, despite issues in the Q-q plots. Or if the findings don't validate using GISTIC this will help avoid spurious findings entering the literature, and the manuscript could proceed with the figure 3 data omitted. Overall the manuscript covers an important topic, so it will be of strong interest to the community even without fig3 data.

Reviewer #2 (Remarks to the Author):

Thank you for the invitation to review the newly updated version of the manuscript by Li et al. "Age Influences on the Molecular Presentation of Tumours".

I still consider the authors did not adequately address my suggestions. I will try to make clearer suggestions:

a. In an article entitled "Age Influences on the Molecular Presentation of Tumours", I would appreciate to read in the abstract something like "Age influences the number of tumor mutations (0.077 per Mb and year) and their evolutionary timing".

b. Please report effect sizes (SNVs/Mb and CNVs/Mb) instead of Spearman correlations. For example: "We also identified positive associations in 11 TCGA, 14 PCAWG, and six AACR GENIE tumour-types (Figure 1A). Of these, nine tumour-types showed consistent results in two of three datasets

(Supplementary Figure 1, Supplementary Table 2) including prostate cancer (TCGA: $p = 0.25$ -> please show here the estimated increase in SNVs/Mb per year instead of p

c. The correct interpretation of a FDR is "the rate of false discoveries". FDR=0.05 translates into "5% of the reported findings are false", which is not acceptable. Please review the complete manuscript highlighting associations with a Bonferroni-corrected p value (or other FWER) smaller than 0.05. Results with FDR<0.05 can be provided as supplementary material, and associations with FDR<0.05 can be briefly described in the article as "potential associations". This major correction would allow the authors to tighten the abstract, as indicated below.

d. - The abstract is not very informative.

Specific mutational signatures (which ones?) are associated with age, reflecting differences in exogenous and endogenous oncogenic processes such as a greater influence of tobacco use in the tumours of younger patients, but more activity of DNA damage repair signatures in those of older patients.

We find that known cancer driver genes (which ones?) are mutated in age-associated frequencies, and these alter the transcriptome and predict for clinical outcomes (which ones?).

We thank the reviewers again for their time and comments. We have made multiple additions and adjustments to the paper, clarified statistical methodology further and:

- Added Pearson's X^2 tests as an orthogonal approach to verify age-associated CNAs
- Added Supplementary Table 2 containing only significant results

Reviewer #1

- The last point (on copy number analysis, points 4 & 5 from the original rebuttal) is more concerning however - as the author acknowledges, the Q-Q plots have some "un-ideal distributions". The $y=x$ line is missing from the plots (and should be added), but from looking at the data the observed p-values are highly inflated compared to what would be expected. This is a concern as it suggests that nearly any gene could come up as a hit in this analysis, which was also an issue from the first rebuttal where thousands of genes were shown to be significantly associated with aging. This is coupled with the concerns of the other reviewer (who is unsure about multiplicity correction). In the latest reply the authors have provided helpful insight into the root cause of the issue, ie that individual SCNA segments span tens or even hundreds/thousands of genes. This is indeed a challenging issue, and in other major studies (eg TCGA) used specialised analysis tools such as GISTIC were used for SCNA gene level analysis (eg see here for a recent example <https://pubmed.ncbi.nlm.nih.gov/32203465/>). I appreciate it's late in the review process to undertake new primary analysis, but if the key findings from figure 3 could be validated using an orthogonal statistical approach that would reassure greatly the readership. For example, if FANCA loss is identified as a significant GISTIC peak in older individuals, but not (or less markedly less strongly) associated in younger individuals, that would show the key findings are valid, despite issues in the Q-q plots. Or if the findings don't validate using GISTIC this will help avoid spurious findings entering the literature, and the manuscript could proceed with the figure 3 data omitted. Overall the manuscript covers an important topic, so it will be of strong interest to the community even without fig3 data.

We fully appreciate the reviewer's concerns that the p-values for a subset of our comparisons show signs of inflation. We note that our methodology here remains quite common in TCGA and PCAWG studies, as an example from the PCAWG bucket of papers, published in this journal:

<https://pubmed.ncbi.nlm.nih.gov/32024823/>

In that study and many others, no assessment is made of Q-Q plots for CNAs because of the challenges raised here. That being said, we definitely appreciate the concern and only note this to reflect that there is no standard in our field for the statistical analysis or presentation of data like this, despite their wide-analysis. The idea of using GISTIC is an interesting one, and as the reviewer recognizes GISTIC is really a *univariate driver discovery tool*. That is a little distinct from what we are doing here: we are not looking at recurrence, but rather associations adjusted for multiple covariates.

To get at the core idea, we have taken an orthogonal, non-parametric statistical approach on discretized age-data. We median-dichotomized age for each tumour type, and tested its association with copy number gains and losses separately using Pearson's X^2 test. We again applied a 10% FDR threshold. Our full statistical procedure is now:

- 1) Univariate assessment with a continuous univariate parametric model (10% FDR)
- 2) Univariate assessment with a discrete univariate non-parametric model (10% FDR)
- 3) The intersection of 1) and 2) are tested with a continuous multivariate model (10% FDR)

We note that the joint statistical stringency here is quite high since we require the intersection of two 10% FDR models, followed by validation of that pool with a third model. When applying this framework, we find 43 tumour-type-specific age-associated CNA drivers (from 58 previously) and 29 pan-cancer CNA drivers (from 32 previously). The major mRNA and survival findings following our analysis of age-associated CNA drivers were all unaffected.

We have made changes to Figure 2, Supplementary tables 5-6, and added Supplementary Table 2. Changes in the text related to this updated analysis are throughout the copy number section “CNA Differences Associated with Transcriptomic Changes” (Lines 315-391) and in particular at:

Lines 324-327

“We further used Pearson’s X^2 tests to evaluate all driver CNAs as an orthogonal measure to minimize false positive hits: we take only results that pass the two stacked 10% FDR thresholds from our statistical framework and the 10% threshold on FDR-adjusted Chi-squared p-values to be significant.”

And in Methods at 704-708

“We also used Chi-squared tests to evaluate all driver CNAs in all tumour-types. We tested the association of gains/losses with median dichotomized age. Significant age-associations must pass the two 10% FDR thresholds from our statistical framework and the 10% threshold on FDR-adjusted Chi-squared p-values. Bonferroni adjusted p-values are also presented in Supplementary Tables 1, 5-6.”

Reviewer #2

Thank you for the invitation to review the newly updated version of the manuscript by Li et al. “Age Influences on the Molecular Presentation of Tumours”. I still consider the authors did not adequately address my suggestions. I will try to make clearer suggestions:

We thank the reviewer for their ongoing suggestions and review of this manuscript.

a. In an article entitled “Age Influences on the Molecular Presentation of Tumours”, I would appreciate to read in the abstract something like “Age influences the number of tumor mutations (0.077 per Mb and year) and their evolutionary timing”.

We have added this to the abstract.

b. Please report effect sizes (SNVs/Mb and CNVs/Mb) instead of Spearman correlations. For example: “We also identified positive associations in 11 TCGA, 14 PCAWG, and six AACR GENIE tumour-types (Figure 1A). Of these, nine tumour-types showed consistent results in two of three datasets (Supplementary Figure 1, Supplementary Table 2) including prostate cancer (TCGA: $\rho = 0.25$ -> please show here the estimated increase in SNVs/Mb per year instead of ρ ”

In addition to the description of all significant estimates of per year increase in SNV density and PGA in **Table 2** (reproduced below), we have included estimates in text where appropriate (underlined):

Lines 115-128:

“There were pan-cancer positive correlations between age and SNV density in TCGA (pan-TCGA: $\rho = 0.31$, FDR-adjusted LNR $p = 4.1 \times 10^{-57}$, Bonferroni-adjusted LNR $p = 4.1 \times 10^{-57}$) and PCAWG ($\rho = 0.43$, FDR-adjusted LNR $p = 1.6 \times 10^{-26}$, Bonferroni-adjusted LNR $p = 4.1 \times 10^{-57}$) data. Using TCGA and PCAWG data, we estimate that SNV density increases at a rate of 0.077 mutations per megabase pair per year (Table 2, Methods). We also identified positive associations in 11 TCGA, 14 PCAWG, and six AACR GENIE tumour-types (**Figure 1A**). Of these, nine tumour-types showed consistent results in two of three datasets (**Supplementary Figure 1, Supplementary Table 2, Supplementary Table 3**) including prostate cancer (TCGA: $\rho = 0.25$, FDR-adjusted LNR $p = 0.015$, Bonferroni-adjusted LNR $p = 0.13$; PCAWG: $\rho = 0.48$, FDR-adjusted LNR $p = 1.2 \times 10^{-4}$, Bonferroni-adjusted LNR $p = 8.7 \times 10^{-4}$, estimated 0.12 mut/Mbp/year; Figure 1B). Estimates for per year increase in mutation density are given in **Table 2** for the nine tumour-types with consistent evidence in at least two datasets.”

Lines 159-178:

“We found that in pan-cancer analysis, PGA increased with age in PCAWG ($\rho = 0.19$, FDR-adjusted LNR $p = 0.022$, Bonferroni-adjusted LNR $p = 0.068$) and AACR GENIE ($\rho = 0.041$, FDR-adjusted LNR $p = 0.050$, Bonferroni-adjusted LNR $p = 0.16$) (**Figure 1D**) and estimate that PGA increases at 0.010% per year (Table 2). We also identified positive correlations in six TCGA, three PCAWG, and three AACR GENIE tumour-types. Again, prostate cancer showed consistent age-PGA associations, this time in all three datasets (TCGA: $\rho = 0.17$, FDR-adjusted LNR $p = 6.7 \times 10^{-5}$, Bonferroni-adjusted LNR $p = 1.8 \times 10^{-4}$; PCAWG: $\rho = 0.27$, FDR-adjusted LNR $p = 3.0 \times 10^{-3}$, Bonferroni-adjusted LNR $p = 4.4 \times 10^{-3}$; AACR GENIE: $\rho = 0.11$, FDR-adjusted LNR $p = 0.050$, Bonferroni-adjusted LNR $p = 0.20$, increase of 0.2%/year; Figure 1E). Age was associated with PGA in stomach cancer data in TCGA with an estimated increase of 0.19% per year ($\rho = 0.11$, FDR-adjusted LNR $p = 0.011$, Bonferroni-adjusted LNR $p = 0.011$) and AACR GENIE ($\rho = 0.38$, FDR-adjusted LNR $p = 0.041$, Bonferroni-adjusted LNR $p = 0.083$), and while other age-PGA correlations were not statistically significant across multiple datasets, they showed similar effect sizes (**Supplementary Figure 1**). Intriguingly, we detected negative age-PGA associations in TCGA lung adenocarcinomas (**Figure 1D**), and correspondingly negative associations in PCAWG ($\rho = -0.13$) and AACR GENIE lung tumours ($\rho = -0.099$) (**Supplementary Figure 1**). Estimates for per year increase in PGA are given in **Table 2** for the five tumour-types with consistent evidence in at least two datasets.”

Table 2:

Tumour-type	ΔMut/Mbp per year (95% CI)	ΔPGA per year (95% CI)
Breast Carcinoma	0.064 (0.029-0.95)	-
Glioblastoma	0.018 (-0.020-0.056)	-
Head and Neck Carcinoma	0.14 (0.071-0.21)	-
Clear Cell Renal Cell Carcinoma	0.018 (0.0084-0.027)	-
Lung Adenocarcinoma	-	-0.17 (-0.24--0.10)
Hepatocellular Carcinoma	0.069 (0.032-0.11)	-
Ovarian Cancer	-	0.61 (0.41-0.81)
Prostate Cancer	0.12 (0.034-0.20)	0.2 (0.13-0.27)
Sarcoma	0.044 (-0.0091-0.098)	-
Stomach Adenocarcinoma	0.31 (0.14-0.49)	0.19 (0.064-0.31)
Thyroid Cancer	0.0082 (0.0066-0.0097)	0.067 (0.036-0.098)
Pan-cancer	0.077 (0.049-0.10)	0.010 (-0.01-0.030)

c. The correct interpretation of a FDR is “the rate of false discoveries”. FDR=0.05 translates into “5% of the reported findings are false”, which is not acceptable. Please review the complete manuscript highlighting associations with a Bonferroni-corrected p value (or other FWER) smaller than 0.05. Results with FDR<0.05 can be provided as supplementary material, and associations with FDR<0.05 can be briefly described in the article as “potential associations”. This major correction would allow the authors to tighten the abstract, as indicated below.

We would first like to continue to gently disagree the statement that “5% of reported findings are false... is not acceptable”. To the contrary, this is absolutely standard in this field and as we have noted is reflected in >100 papers from TCGA and ICGC projects. In fact, interpretation of our work in the context of the prior literature will actually be more challenging by applying statistical thresholds that are out of line with the convention in this field.

We also note that our statistical threshold estimates are perhaps more stringent than the reviewer recognizes. All reported hits in our study now have met our threshold in three distinct statistical tests: a univariate continuous parametric test, a univariate discrete non-parametric test and a multivariate continuous parametric test. This is again more stringent than the standard threshold that is typically used in the majority of hypothesis-testing statistical analyses. We are reporting raw p-values to allow readers to interpret the unadjusted results, and Bonferroni-significant values are highlighted in the text as requested.

We appreciate that our description of our statistical methods may have been unclear and have added description at Lines 93-98

“We perform two rounds of multiple testing adjustment: once at the first univariate stage, and again at the second multivariate stage, both using the Benjamini-Hochberg false discovery rate (FDR) procedure. Our findings must pass stacked FDR thresholds of 10% on top of 10% after both stages of analysis, representing a stringent combined threshold of 1%. These FDR-adjusted p-values are used throughout.”

And in Methods at Lines 629-632:

“FDR adjustment was performed for p-values for the age variable significance estimate and an FDR threshold of 10% was used to determine statistical significance. Statistically significant findings must therefore pass two rounds of FDR-adjustment, one at the univariate stage and the second at the multivariate stage.

d. - The abstract is not very informative.

Specific mutational signatures (which ones?) are associated with age, reflecting differences in exogenous and endogenous oncogenic processes such as a greater influence of tobacco use in the tumours of younger patients, but more activity of DNA damage repair signatures in those of older patients.

We find that known cancer driver genes (which ones?) are mutated in age-associated frequencies, and these alter the transcriptome and predict for clinical outcomes (which ones?).

Our abstract now incorporates many of these comments, although of course the word-count limits adding extensive detail in some areas. It now reads (with some of the key concerns raised by the reviewer **emphasized**).

“Cancer is often called a disease of aging. There are numerous ways in which cancer epidemiology and behaviour change with the age of the patient. The molecular bases for these relationships remain largely underexplored. To characterize them, we analyzed age-associations in the nuclear and mitochondrial somatic mutational landscape of 20,033 tumours across 35 tumour-types. Age influences both the number of mutations in a tumour (**0.077 mutations/Mbp per year**) and their evolutionary timing. Specific mutational signatures are associated with age, reflecting differences in exogenous and endogenous oncogenic processes such as **a greater influence of tobacco use** in the tumours of younger patients, but more activity of **DNA damage repair signatures** in those of older patients. We find that known cancer driver genes such as **CDKN2A** and **CREBBP** are mutated in age-associated frequencies, and these alter the transcriptome and predict for clinical outcomes. These effects are most striking in brain cancers where alterations like **SUFU** loss and **ATRX** mutation are age-dependent prognostic biomarkers. Using three cancer datasets, we show that age shapes the somatic mutational landscape of cancer, with clinical implications.”

REVIEWERS' COMMENTS

Reviewer #1 (Remarks to the Author):

I have re-reviewed this latest set of comments, and while I would really like to support this paper for publication (the topic is of high interest), the remaining issue is still unfortunately not adequately addressed. In summary:

- The remaining issue relates to copy number analysis (original points 4 & 5). In the latest rebuttal the authors have implemented extra rounds of testing, i.e. performed 3 rounds of significance testing, to bring down the number of significant associations. But really the underlying issue with incorrect statistical methods has not been addressed, and hence there is a strong risk they have just filtered down the results so it's less extreme, but still the results are incorrect. I agree with the authors that there is no standard approach for this type of analysis in the field, but I have never seen other papers where thousands of genes are reported as significant (as it was presented in the first draft of the paper). So reapplying methods from other papers is not really appropriate, if in your particular research question they generate thousands of potential false-positives. It's good this number has been reduced, but still it's not clear if the results are correct or not. Also, the level of validation looks very poor, e.g. in figure 3A, 20 significant events are identified in the discovery series, of which only 1 validates (FANCA). Regarding the authors comments on GISTIC, it is not correct that confounders cannot be controlled for, please see methods in the previously highlighted paper <https://www.ncbi.nlm.nih.gov/pmc/articles/PMC7136154/>: "In order to identify brain metastatic drivers, we performed a case-control analysis on the frequencies of copy-number aberrations (Extended Data Fig. 5). Total copy-number segments produced by ABSOLUTE (v1.4)48 from the case and control cohorts were independently analyzed by GISTIC51. To account for confounding covariates, the segment profiles of control samples were multiplied by the matching weights (see "Control cohort and matching")." So GISTIC could be implemented with covariates, using the above method and the discretized groups the authors have already defined – e.g. an analysis of young vs old individuals. I understand however that implementing a new method at this late stage is not likely what the authors want to do, and I feel bad that so much discussion has ensued on this work, but my comment are just purely out of concern for the correctness of the results.

In terms of next steps, either i) the above GISTIC method should be implemented, which is probably the most appropriate state of the art method for this question, or ii) the copy number results could be scaled right back (e.g. take out fig.3, or put it all in supplementary, except results which validate).

Reviewer #2 (Remarks to the Author):

I have no further comments.

Reviewer #4 (Remarks to the Author):

No comments to authors

Reviewer #1

I have re-reviewed this latest set of comments, and while I would really like to support this paper for publication (the topic is of high interest), the remaining issue is still unfortunately not adequately addressed. In summary:

- The remaining issue relates to copy number analysis (original points 4 & 5). In the latest rebuttal the authors have implemented extra rounds of testing, i.e. performed 3 rounds of significance testing, to bring down the number of significant associations. But really the underlying issue with incorrect statistical methods has not been addressed, and hence there is a strong risk they have just filtered down the results so it's less extreme, but still the results are incorrect. I agree with the authors that there is no standard approach for this type of analysis in the field, but I have never seen other papers where thousands of genes are reported as significant (as it was presented in the first draft of the paper). So reapplying methods from other papers is not really appropriate, if in your particular research question they generate thousands of potential false-positives. It's good this number has been reduced, but still it's not clear if the results are correct or not.

Also, the level of validation looks very poor, e.g. in figure 3A, 20 significant events are identified in the discovery series, of which only 1 validates (FANCA). Regarding the authors comments on GISTIC, it is not correct that confounders cannot be controlled for, please see methods in the previously highlighted paper <https://www.ncbi.nlm.nih.gov/pmc/articles/PMC7136154/>: "In order to identify brain metastatic drivers, we performed a case-control analysis on the frequencies of copy-number aberrations (Extended Data Fig. 5). Total copy-number segments produced by ABSOLUTE (v1.4)48 from the case and control cohorts were independently analyzed by GISTIC51. To account for confounding covariates, the segment profiles of control samples were multiplied by the matching weights (see "Control cohort and matching")." So GISTIC could be implemented with covariates, using the above method and the discretized groups the authors have already defined – e.g. an analysis of young vs old individuals. I understand however that implementing a new method at this late stage is not likely what the authors want to do, and I feel bad that so much discussion has ensued on this work, but my comment are just purely out of concern for the correctness of the results.

In terms of next steps, either i) the above GISTIC method should be implemented, which is probably the most appropriate state of the art method for this question, or ii) the copy number results could be scaled right back (e.g. take out fig.3, or put it all in supplementary, except results which validate).

We thank the reviewer for their time and comments. We appreciate the reviewer's concerns on our CNA analysis and believe the changes we have made with the reviewer's guidance have strengthened the section. The core change from our first submission is moving from a genome-wide CNA analysis to focus on CNA drivers. CNAs are large structural events affecting hundreds or thousands of genes simultaneously, leading to highly similar CNA profiles for adjacent genes: this contributed to the large number of results we presented in our first submission. As they recommend, we have significantly scaled-back the copy number results as suggested. We have reduced the text in the CNA section to focus on key findings, and modified Figure 3 to five panels from six. Our CNA-related supplementary materials are unchanged from the last submission and present the full results of all relevant analyses.

Shifting our analysis to driver CNAs allowed us to focus on those genes that are thought to be targeted by these CNAs. Driver CNAs confer a selective advantage to tumour cells, compared with the hundreds of passenger alterations that do not. However, we did not perform driver discovery in this study. Instead, we leveraged existing knowledge and databases compiled by the genomics community through the application of tools such as GISTIC2.0 to curate a list of driver losses and gains. We analysed this subset of driver CNAs using univariable and multivariable analysis to control for confounding variables, and performed multiple testing adjustment at each stage. Applying GISTIC2.0 in the manner described in <https://www.ncbi.nlm.nih.gov/pmc/articles/PMC7136154/> may allow us to perform differential driver discovery by age, but we believe the methods in our study are comparable and equally valid. We have significantly scaled back our CNA section to focus only on the key findings. We have also condensed Figure 3, and it includes only findings that are significant in at least one dataset and supported by similar effect sizes in at least one other, as recommended. In addition, all CNA results are presented in Supplementary Data. We sincerely thank the reviewer for their thoughtful suggestions.